# Geometry-Aware Metric for Dataset Diversity via Persistence Landscapes

## Abstract

Diversity can be broadly defined as the presence of meaningful variation across elements, which may be viewed from multiple perspectives, including statistical variation and geometric structural richness in the dataset. Existing diversity metrics, such as feature-space dispersion and metric-space magnitude, primarily capture distributional variation or entropy, while largely neglecting the geometric structure of datasets. To address this gap, we introduce a framework based on topological data analysis (TDA) and persistence landscapes (PLs) to extract and quantify geometric features from data. This approach provides a theoretically grounded means of measuring diversity beyond entropy, capturing the rich geometric and structural properties of datasets. Through extensive experiments across diverse modalities, we demonstrate that our proposed PLs-based metric (PLDiv) is powerful, flexible, and interpretable, directly linking data diversity to its underlying geometry and offering new insights for dataset construction, augmentation, and evaluation.

## 1 Introduction

Life itself depends on diversity, as an ecosystem may collapse when a few species vanish, yet a single new species may reshape balance by either enriching resilience or triggering instability. In machine learning and artificial intelligence, data diversity plays the same essential role. Studying diversity has long been a central concern at nearly every stage of ML/AI: from data collection to ensure representational balance, to data and model evaluation for fairness and robustness (Rolf et al., 2021; Clemmensen & Kjærsgaard, 2022; Kim et al., 2025), to model training where variation prevents overfitting, and to model generalization, where data diversity reduces the gap between training distributions and real-world deployment (Liu & Zeldes, 2023; Ortega et al., 2022; Yu et al., 2022; Bian & Chen, 2021; Wang et al., 2020). It is well known that exposure to a wide range of data structures, styles, and semantic patterns supports the learning of more abstract, transferable representations, allowing for more capable and resilient models (Rebuffi et al., 2021; Shorten & Khoshgoftaar, 2019; Zhang, 2017). Recent work further demonstrates that diversity in training data influences the weight matrices of neural networks, directly affecting both in-distribution and out-of-distribution performance (Ba et al., 2024).

Yet beyond performance, a newer—and arguably more urgent—motivation for us to study diversity is the need to confront a growing risk. Today's generative models are trained on overlapping, internet-scale corpora, then reused and adapted across countless applications. As these models are increasingly integrated into real-world writing, content creation, visual and audio materials, and codes, their outputs feed back into the very data streams that will train the next generation of models. Recent studies show that alignment-tuned models such as InstructGPT already exhibit significant reductions in lexical and conceptual diversity (Padmakumar & He, 2023). Unlike traditional data limitations, this homogenization is self-reinforcing: models trained on uniform outputs further reinforce uniformity in subsequent models (Bertrand et al., 2023; Alemohammad et al., 2024). The danger is not limited to text generation, as the same internet-scale sources, standardized pipelines, and optimization objectives underpin generative models across all data modalities. Combined with algorithmic feedback loops, platform-driven content shaping, and widespread reuse of foundation models, these forces may steadily contract the expressiveness and conceptual space of generative AI at scale.

At this stage, diversity is no longer just a desirable property; it has become a boundary condition for innovation, adaptability, and human-centered AI design. Meeting this challenge requires us to understand what *real* diversity is and be able to measure it. Reliable measurement allows us not only to detect the narrowing trajectories of generative models, but also to design interventions that can preserve and promote diversity. This understanding, in turn, can guide future efforts toward diversity-aware data collection, synthetic data generation, data augmentation strategies, and dataset–task alignment.

To quantify diversity, metrics such as the Vendi Score (Dan Friedman & Dieng, 2023) have been introduced, drawing inspiration from "community diversity" in ecology and biology (Daly et al., 2018; Leinster, 2021). Recently, measures based on magnitude (Limbeck et al., 2024) and probability-distribution views of similarity matrices (Zhu et al., 2025) have also been proposed. These methods are valuable, but none of them genuinely considers data from a geometric perspective, even when they claim to capture some geometric information.

We envision a deeper link between the geometric structure of data and its diversity. For instance, as a fundamental geometric property, curvature is inherently linked to diversity (Limbeck et al., 2024): positive curvature, as on a sphere, compresses points and restricts possible configurations, while negative curvature, as in hyperbolic geometry, spreads space out faster, enabling richer variation. Topological data analysis (TDA) provides tools to capture the shape of data, encoding its structural geometry. By recognizing the connection between the persistent homology (PH) merging process (Edelsbrunner et al., 2002; 2008) and agglomerative hierarchical clustering (Murtagh & Contreras, 2012), we employ a vectorized representation of PH called the persistence landscapes (PLs) to estimate diversity. We compute the cumulative integral of their tent functions, which is referred to as persistence landscapes-based diversity (PLDiv), as shown in Fig. 1. PLDiv has a clear intuition, strong theoretical support, and interpretable results.

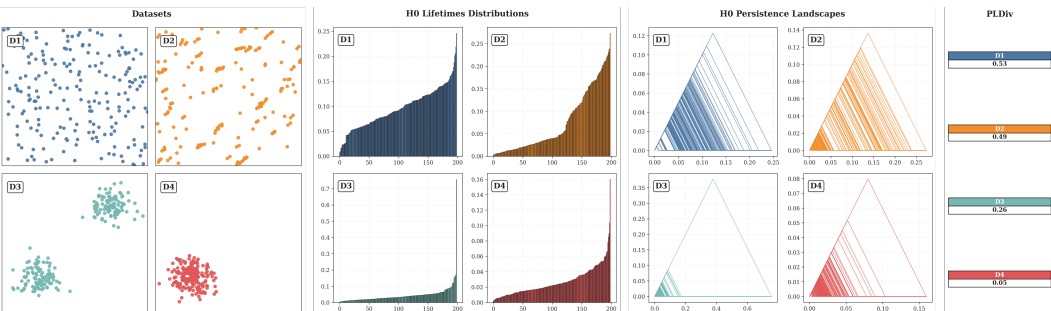

Figure 1: Illustration of PLDiv on four synthetic datasets. D1: uniformly scattered points; D2: less evenly spread distribution; D3: two separated clusters; D4: a single compact cluster with minimal diversity. We extract $H_0$ features via persistent homology, where lifetimes measure how long clusters persist before merging with their closest neighbors. Persistence landscapes capture these patterns, and PLDiv, defined as the sum of their integrals, reflects both scale and persistence, aligning with the datasets' decreasing diversity.

Our contributions are summarized as follows:

- We propose a persistence landscape-based diversity measure, PLDiv. The core idea is that persistence homology encodes geometric information; thus, PLDiv highlights the value of topological features that play a key role in capturing meaningful structural patterns.

- We establish the theoretical foundations of PLDiv by proving that it satisfies multiple diversity axioms postulated by Leinster & Cobbold (2012), thereby ensuring its interpretability and principled behavior.

- Through comprehensive experiments across various tasks and data modalities, we demonstrate that PLDiv captures geometrical and structural diversity more effectively than conventional entropy-based approaches and offers practical advantages in robustness and interpretability.

To the best of our knowledge, we are the first to apply TDA concepts to measuring data diversity. Our study provides a novel approach to data diversity measurement and offers both the theoretical foundation and interpretability for a data geometry-aware diversity measure.

## 2 RELATED WORK

### 2.1 DIVERSITY MEASUREMENT

Several reference-based metrics compare generated data with human or gold-standard corpora. The Fréchet Inception Distance (FID) (Heusel et al., 2017) and related Inception Score were among the first to use pretrained embeddings to measure alignment between real and synthetic data distributions. More recently, MAUVE (Pillutla et al., 2021) quantified distributional gaps between model and human text, while precision–recall metrics (Kynkäänniemi et al., 2019; Bronnec et al., 2024) provided a decomposition into fidelity (precision) and diversity (recall). Extensions such as density and coverage metrics (Naeem et al., 2020) improved robustness against outliers and unstable density estimates. Nevertheless, these methods are fundamentally tied to reference datasets, often entangle fidelity with diversity, and remain sensitive to embedding choices or manifold approximations.

A different line of work has explored representation-level measures that aim to be reference-free. Early proposals such as diversity, density, and homogeneity Lai et al. (2020) assessed dispersion in embedding spaces, but they remained limited to simple distributional statistics. More principled approaches emerged with entropy- or kernel-based methods: the Vendi Score (Dan Friedman & Dieng, 2023) measures diversity as the exponential of Shannon entropy derived from the similarity spectrum, while Renyi Kernel Entropy (RKE) and its variant RRKE (Jalali et al., 2023) extend this perspective using quantum information theory. However, such approaches often require expensive eigenvalue or singular-value decompositions, limiting their scalability to large datasets. Building on efficiency and separability, DCScore (Zhu et al., 2025) reframes diversity measurement as a classification problem, avoiding eigenvalue computations and yielding faster, more scalable estimates. Complementary to this, magnitude-based methods (Limbeck et al., 2024) quantify effective dataset size across scales, offering metrics such as MAGAREA (reference-free) and MAGDIFF (reference-based). While these methods provide multi-scale summaries, they depend on tuning scale parameters and still abstract away the geometric or topological structures that can differentiate datasets with the same dispersion.

### 2.2 PERSISTENT HOMOLOGY

Persistent Homology (PH) (Edelsbrunner et al., 2002; 2008) is a central tool in TDA for uncovering the underlying shape of data, typically represented as point clouds. By constructing nested simplicial complexes across scales and applying homology, PH tracks the birth and death of topological features such as connected components, loops, and voids. The result is a multi-scale summary, often visualized as barcodes or persistence diagrams, which distinguishes significant long-lived features from noise and is provably stable to perturbations.

Building on these foundations, subsequent efforts have explored scalar invariants and geometric inference from persistence. Govc & Hepworth (2021) introduced persistent magnitude, a signed, exponentially weighted sum over barcode intervals that refines classical magnitude theory. This approach provides interpretable scalar summaries encoding geometric complexity, including curvature, but it compresses the full topological signature into a single number, limiting its ability to capture heterogeneity or higher-order organization. In parallel, Bubenik et al. (2020) demonstrated that persistence can recover curvature information from sampled manifolds by combining diagrams with persistence landscapes, showing that even short-lived features carry meaningful geometric signals. While powerful, this line of work primarily targets smooth continuous geometry rather than irregular or combinatorial variation common in real-world datasets. Together, these directions underscore the expressive capacity of PH, yet also highlight an open gap: existing uses either oversimplify persistence or focus narrowly on geometric inference, leaving the systematic role of PH in quantifying dataset diversity underexplored.

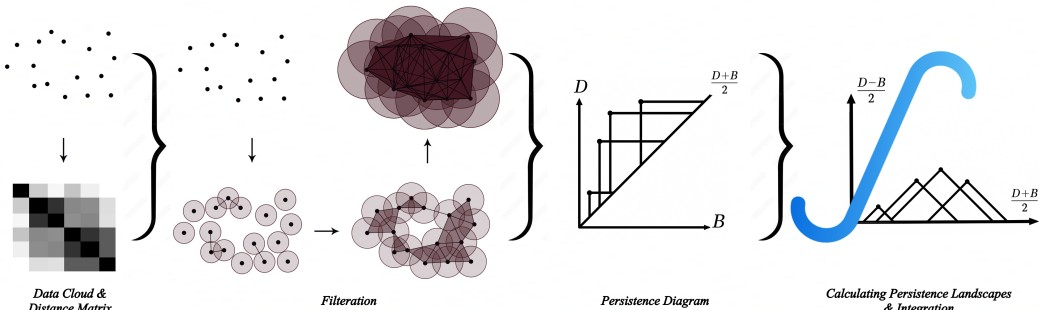

*Data Cloud &
Distance Matrix*      *Filteration*      *Persistence Diagram*      *Calculating Persistence Landscapes
& Integration*

Figure 2: Illustration of the PLDiv pipeline. Using a data cloud or its distance matrix, we build a filtration of simplicial complexes and track the birth and death of $H_0$ components by persistent homology. The resulting persistence diagram is then used to calculate persistence landscapes. Lastly, PLDiv is obtained by integrating these landscapes and provides a metric for the dataset diversity.

## 3 PRELIMINARIES

### 3.1 PERSISTENCE DIAGRAMS

PH provides a multiscale description of the topological structure of data. Starting from a point cloud $\mathcal{X} = \{x_1, \ldots, x_n\}$, it builds a nested sequence of simplicial complexes (a filtration), such as the Vietoris–Rips filtration. This filtration can be understood as growing balls (or "bubbles") of radius $r$ around each data point and increasing $r$ gradually. As the radius grows, the bubbles begin to overlap, creating higher-dimensional simplices (see Fig. 2). In this process, new topological features such as connected components, loops, and voids appear and eventually vanish when the bubbles merge or fill in. This viewpoint highlights that persistent homology captures how the topology of the data evolves across scales of the underlying radius parameter.

Formally, each topological feature is associated with a birth time $b_i$, the smallest radius at which it appears, and a death time $d_i$, the radius at which it disappears (for instance, when two connected components merge or when a loop becomes filled). The difference $\ell_i = d_i - b_i$ is called the *lifetime* (or persistence) of the feature and quantifies its robustness across scales.

The output of persistent homology is summarized in a *persistence diagram*, defined as the multiset

$$\mathcal{D} = \{(b_i, d_i)\}_{i=1}^m, \quad b_i < d_i,$$

where each point $(b_i, d_i)$ represents the birth and death scales of a feature. The diagram is typically plotted in the plane $\mathbb{R}^2$, with each feature as a point above the diagonal $b = d$. Features with long lifetimes (points far from the diagonal) are often interpreted as meaningful structural signals in the data, while short-lived features (points near the diagonal) are commonly attributed to noise. Persistence diagrams thus provide a compact and interpretable summary of the multiscale topological properties of the dataset.

### 3.2 PERSISTENCE LANDSCAPES

Although persistence diagrams provide a geometric summary of topological features, they are multisets, represented by points on a plane, which makes it challenging to apply classical statistical and machine learning techniques directly. To address this problem, Bubenik et al. (2015) introduced *persistence landscapes*, a functional summary of persistent homology that embeds the information of a persistence diagram into a Banach space, enabling the use of standard statistical tools.

Given a persistence diagram $\mathcal{D} = \{(b_i, d_i)\}_{i=1}^m$, we first associate each birth-death pair $(b_i, d_i)$ with a piecewise linear "tent" function.

$$\lambda_{(b,d)}(t) = \begin{cases} t - b, & b \leq t \leq \frac{b+d}{2}, \\ d - t, & \frac{b+d}{2} < t \leq d, \\ 0, & \text{otherwise.} \end{cases}$$

This function attains its maximum value, $\frac{d_i - b_i}{2}$, at the midpoint of the interval. The persistence landscape is then defined as the sequence of functions

$$\lambda_k(t) = k\text{-th largest value among } \{\lambda_{(b_i,d_i)}(t)\}_{i=1}^m, \quad k = 1, 2, \ldots$$

for each $t \in \mathbb{R}$. Thus, $\lambda_1$ records the largest "tent" value at each $t$, $\lambda_2$ records the second largest, and so forth. Collectively, the functions $\{\lambda_k\}_{k \geq 1}$ constitute the persistence landscape.

Persistence landscapes inherit stability from persistence diagrams and have the advantage of lying in the $L^p$ function space. The persistence landscape is a vectorized form of a persistence diagram, equivalent to a 45° rotation that preserves all information, with X = $(d + b)/2$ and Y = $(d - b)/2$ (see Fig. 2).

## 4 METHODOLOGY

### 4.1 DIVERSITY MEASURE VIA PERSISTENCE LANDSCAPES

**Definition 4.1.** Let $\mathcal{X} = \{x_1, \ldots, x_n\}$ be a dataset and let $\Lambda(\mathcal{X}) = \{\lambda_k\}_{k \geq 1}$ denote its persistence landscape obtained from persistent homology. The *persistence landscapes based diversity* score, PLDiv($\mathcal{X}$), is defined as

$$\text{PLDiv}(\mathcal{X}) = \sum_{k=1}^{\infty} \int_{\mathbb{R}} \lambda_k(t) \, dt. \tag{1}$$

The summation is typically finite, as only a finite number of $\lambda_k$ terms are actually non-zero. PLDiv($\mathcal{X}$) measures the cumulative "area under the triangles" of the persistence landscape and quantifies the richness of topological features across all scales.

**Proposition 4.2.** A closed form of PLDiv can be derived. Let $\mathcal{D} = \{(b_i, d_i)\}_{i=1}^m$ be the set of birth–death pairs produced by persistence homology, then

$$\text{PLDiv}(\mathcal{X}) = \sum_{k=1}^{\infty} \int_{\mathbb{R}} \lambda_k(t) \, dt = \sum_{i=1}^{m} \int_{\mathbb{R}} \lambda_{(b_i,d_i)}(t) \, dt = \frac{1}{4} \sum_{i=1}^{m} (d_i - b_i)^2.$$

*Proof.* Each tent function with its supports on the interval $[b_i, d_i]$ is a symmetric isosceles triangle of base length $d_i - b_i$ and height $(d_i - b_i)/2$, hence its area is

$$\int_{\mathbb{R}} \lambda_{(b_i,d_i)}(t) \, dt = \tfrac{1}{2} \cdot (d_i - b_i) \cdot \tfrac{d_i - b_i}{2} = \frac{(d_i - b_i)^2}{4}.$$

Summing them yields the closed form above. We provide a detailed proof in Appendix C.

**Remark 4.3.** The area under $\lambda_k$ measures both the *scale* and the *persistence* of topological features, representing how long and how strongly features persist across scales. Summing across $k$ aggregates contributions across all topological structures, capturing both *local fluctuations* (short lifetimes) and *global connectivity* (long lifetimes).

**Remark 4.4.** A large PLDiv($\mathcal{X}$) indicates that features such as clusters or loops are well-separated and persist across scales, reflecting high structural diversity. Conversely, a smaller value corresponds to a dataset where data points collapse quickly into clusters, eliminating persistent features. In particular, by Proposition 4.2, PLDiv ($\mathcal{X}$) coincides with the second moment of lifetimes of topological features, up to scaling.

**Remark 4.5.** Since the persistence landscape lies in $L^p(\mathbb{R})$, the integral $\int_{\mathbb{R}} \lambda_k(t) \, dt$ can be interpreted as the "expected persistence" of the $k$-th most prominent feature across random scales $t$. From the probabilistic perspective, PLDiv($\mathcal{X}$) represents the total expected persistence across all topological features, analogous to computing an energy functional over the data manifold.

PLDiv($\mathcal{X}$) should be understood as a holistic measure of dataset complexity. Unlike conventional approaches in topological data analysis that treat short-lived features as noise, this measure incorporates the full spectrum of topological features, emphasizing that both long- and short-lived structures contribute to the geometry of the data (follows the insights in Turkes et al. (2022)). In this sense, PLDiv($\mathcal{X}$) provides a unified framework that balances mathematical rigor with interpretability.

In practice, there are many choices for the filtration and the degree of persistent homology. For most tasks, 0-dimensional persistent homology is sufficient, because it efficiently captures the connectivity structure of the dataset while keeping computational costs low. Therefore, our metric (PLDiv) is computed based on $H_0$ features in the following experiments.

## 4.2 AXIOMATIC PROPERTIES OF DIVERSITY

Among core diversity axiomatic properties provided by Leinster & Cobbold (2012) and Leinster (2021), our proposed diversity measure, PLDiv, satisfies four fundamental axioms: effective size, monotonicity, twin property, and symmetry. These axioms provide a foundation for reasonable and robust diversity evaluation. A description of these axioms is provided below, while the formal proofs of these properties on PLDiv are presented in Appendix C.

- **Effective size.** For a fixed number of points, $\text{PLDiv}(\mathcal{X})$ increases when data points are well-separated and decreases as they cluster, reaching a maximum when all points are distinct and a minimum when all are identical.

- **Monotonicity.** Decreasing similarity increases diversity. Fix $n$ and let $\mathcal{X}$ be a point cloud in a metric space. If all pairwise distances in $\mathcal{X}$ are scaled by a factor $\alpha > 1$ (i.e. replace the metric $d(\cdot, \cdot)$ by $\alpha d(\cdot, \cdot)$), then

$$\text{PLDiv}(\alpha \mathcal{X}) > \alpha^2 \text{PLDiv}(\mathcal{X}) \quad \text{if } \alpha > 1, \text{ and vice versa.}$$

- **Twin property.** Adding an exact duplicate of a point does not change $\text{PLDiv}(\mathcal{X})$. The duplicate induces a trivial birth–death pair $(0, 0)$, contributing zero to the diversity score. Let $\mathcal{X}$ be a dataset and let $x_i \in \mathcal{X}$. For the set $\mathcal{X}' = \mathcal{X} \cup \{x_n\}$ where $x_n = x_i$, the diversity is unchanged:
$$\text{PLDiv}(\mathcal{X}') = \text{PLDiv}(\mathcal{X}).$$

- **Symmetry.** PLDiv is invariant to the ordering of data points (permutation invariance). Since persistent homology depends only on the metric structure of $\mathcal{X}$ and $\text{PLDiv}(\mathcal{X})$ is computed from the multiset of intervals $\{(b_i, d_i)\}$, relabeling or reordering points does not affect the value of the score. Let $\mathcal{X} = (x_1, \ldots, x_n)$ be an ordered sequence of points and let $\pi$ be any permutation of $\{1, \ldots, n\}$. For the permuted sequence $\mathcal{X}_\pi = (x_{\pi(1)}, \ldots, x_{\pi(n)})$, we have
$$\text{PLDiv}(\mathcal{X}_\pi) = \text{PLDiv}(\mathcal{X}).$$

## 5 EXPERIMENT & ANALYSIS

### 5.1 CAPTURING DIVERSITY IN SUBSET SELECTION

A long-standing challenge in diversity measurement is the absence of ground truth labels. The issue is especially significant for complex data modalities such as text, where objective evaluation is difficult. To validate our diversity measure, we use outputs of a Determinantal Point Process (DPP) (Kulesza et al., 2012), a probabilistic model that favors selecting diverse subsets from a larger set. Instead of treating all subsets equally, DPP picks those where the elements are dissimilar to one another. Specifically, it works by first measuring the similarity between every pair of points in the dataset using a kernel. Subsets that contain points that are very similar to each other are less likely to be chosen, while subsets with points that are more distinct are more likely. This guarantees that DPP produces a diverse subset, making it particularly effective as ground truth for evaluating data diversity.

We apply k-DPP (Kulesza & Taskar, 2011) (selecting $k$ diverse samples from the entire set) to both a simulation and the ArXiv-10 dataset (Farhangi et al., 2022). In the simulation, we construct a dataset of 200 points arranged into two adjacent clusters, with 100 points per cluster, from which 30 data points are selected. Additionally, we sample 100 data points from the first 1,000 instances of the ArXiv dataset and vectorize them using the text embedding model "all-MiniLM-L6-v2". In both experiments, we use both uniform random sampling and k-DPP for comparison, using the Radial Basis Function (RBF) kernel for the simulation and cosine similarity for the similarity matrix construction in DPP for the ArXiv dataset. As shown in Fig. 3, our metric PLDiv effectively quantifies the higher

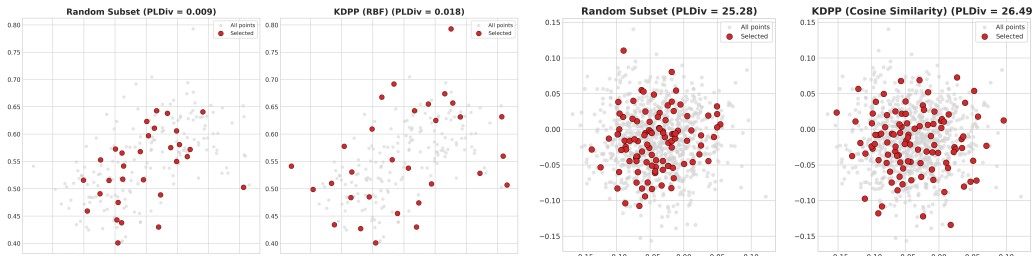

Figure 3: k-DPP selects a $k$-diverse subset from the entire dataset. The two plots on the left present results from simulated data: the first one shows random sampling, while the second one shows k-DPP. The two plots on the right correspond to the ArXiv dataset, with the first one showing random sampling and the second one showing k-DPP. Data points selected by KDPP are scattered more diversely compared to random sampling. PLDiv can successfully capture these subtle differences.

diversity of the DPP-sampled subset compared to the random one, demonstrating its effectiveness. This suggests that PLDiv effectively captures diversity in the metric space, reflecting even small variations and making it well-suited for comparing data diversity across different datasets.

## 5.2 CHARACTERIZING GEOMETRY WITH CURVATURE

As a fundamental property in geometry, curvature quantifies the extent to which a manifold deviates from being flat, thereby governing the behavior of distances within that space. Curvature inherently relates to diversity (Limbeck et al., 2024): On positively curved spaces, such as spheres, data points concentrate and the variety of configurations is reduced; while on negatively curved spaces, such as hyperbolic disks, distances spread apart more quickly, creating a greater range of possible arrangements. Being able to recover curvature from point clouds offers a principled way to validate whether a diversity measure is geometry-aware, rather than relying solely on pairwise dissimilarities. This is important because modern representation learning often places data in non-Euclidean spaces, such as spherical or hyperbolic embeddings, where curvature plays a key role in structuring similarity. A diversity measure sensitive to curvature ensures better representation of the data manifold's geometry.

To this end, we compare PLDiv against several established metrics, including Vendi Score, DCScore, and MAGAREA on the dataset (Turkes et al., 2022), by computing similarity scores from the data and using these scores as features to regress the curvature labels. We employ an SVR (support vector regression) model with an RBF kernel and perform 5-fold cross-validation. For Vendi Score and DCScore, we consider both L1 distance and RBF as similarity functions, whereas MAGAREA uses the default Euclidean distance. Table 1 indicates that the performance of other metrics, such as Vendi

Table 1: PLDiv estimates curvature

| Method | MSE ($\downarrow$) |
|---|---|
| SVR(Vendi Score, L1 kernel) | $0.229 \pm 0.042$ |
| SVR(Vendi Score, RBF kernel) | $0.053 \pm 0.004$ |
| SVR(DCScore, L1 kernel) | $0.134 \pm 0.019$ |
| SVR(DCScore, RBF kernel) | $0.052 \pm 0.004$ |
| SVR(MAGAREA, Euclidean) | $0.120 \pm 0.010$ |
| **SVR(PLDiv)** | $\mathbf{0.039 \pm 0.001}$ |
| **SVR(Sparse PLDiv)** | $\mathbf{0.040 \pm 0.001}$ |

Score and DCScore, is highly dependent on the choice of similarity functions, and PLDiv is the strongest predictor for capturing data geometric structure. The Sparse PLDiv uses the sparse Rips filtration to reduce computation efforts (see Section 5.6).

## 5.3 SEMANTIC DIVERSITY IN TEXT EMBEDDINGS

We investigate the utility of PLDiv as a measure of semantic diversity encoded in text embeddings. We use the dataset from Tevet & Berant (2021), which contains 1,000 sets of 10 sentences generated from unique prompts across three distinct tasks: story completion (story), dialogue response generation (resp), and three-word prompt completion (prompt). For each prompt, 10 candidate outputs were produced by varying the softmax temperature $dec$, resulting in a dataset comprising 1,000

prompts, each associated with 10 output sentences. Subsequently, human evaluators annotated a subset of 200 prompts, with 10 responses per prompt, to obtain the mean human evaluation score (*ABS-HDS*), forming the human dataset. $Dec$ demonstrates the trade-off between quality and diversity in text generation, as lower temperatures increase fidelity by discouraging low-probability tokens, but at the cost of diversity in sampling. *ABS-HDS* serves as the ground truth reflecting how humans perceive text diversity. Accordingly, we use linear regression with 5-fold cross-validation to analyze the relationship between response diversity measurements and temperature settings (as a proxy for diversity in the $dec$ dataset) or the human diversity scores (in the *ABS-HDS* dataset), assessed using $R^2$ and MSE. In addition, we compute Pearson's correlation and perform 1,000 bootstrap iterations to derive confidence intervals. Each response set is embedded using five models: "bert-large-nli-stsb-mean-tokens", "all-MiniLM-L12-v2", and "all-mpnet-base-v2", "Qwen3-Embedding-4B", and "Qwen3-Embedding-8B".

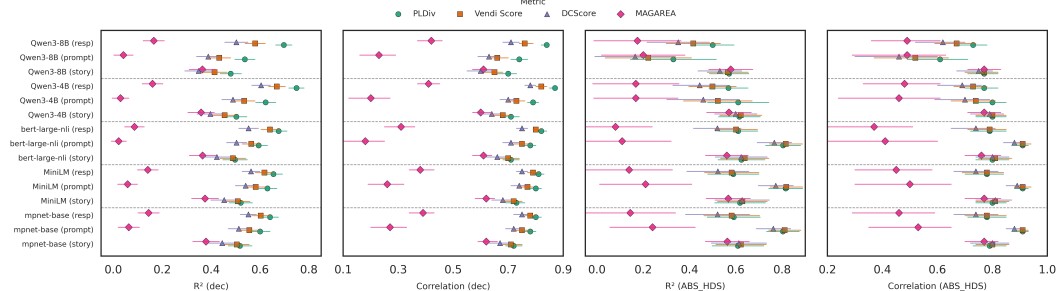

Figure 4: Demonstration that PLDiv achieves superior performance over alternative diversity metrics in predicting ground-truth diversity across tasks and embedding models. Points with different shapes denote to mean $R^2$ and correlation scores, with error bars indicating standard deviations across 5 repeated cross-validation trials. Experiments with *ABS-HDS* exhibit larger error bars due to its smaller sample size.

Fig. 4 visualizes the $R^2$ and correlation results across all tasks and embedding models. PLDiv consistently outperforms all other metrics across tasks and embedding models in temperature-based evaluations. It also demonstrates superior performance in dialogue response generation across all models, as well as in evaluations on two recent embedding models (Qwen3-4B and Qwen3-8B) for all tasks assessed by human judgments. Moreover, PLDiv performs comparably to the Vendi Score in both story completion tasks and prompt tasks for human evaluations, while outperforming DCScore and MagArea. Detailed MSE results and performance analyses under different distance matrix settings are provided in Appendix D.4. Overall, these results demonstrate that PLDiv effectively captures the semantic diversity encoded in text embeddings.

## 5.4 DIVERSITY EVALUATION FOR IMAGE EMBEDDINGS

To demonstrate PLDiv's efficacy for image dataset evaluation, we tested it on Colored MNIST (Deng, 2012). Following the methodology of Ospanov et al. (2024), the number of labels served as the ground truth for diversity, where a higher label count signifies a more diverse set. Comparisons are conducted against Vendi Score, Magnitude, and DCScore, using two embedding models: Inception V3 and ResNet-18. Starting with a single class, we iteratively add one class at a time based on the previous data until all 10 classes are included. To facilitate a direct comparison, each metric was subsequently normalized to the [0, 1] interval (Min–max). This linear transformation preserves the underlying trends and the correlation of each score against the number of classes present in the evaluation.

In Fig. 5, both PLDiv and MAGAREA exhibit a consistent and reliable correlation with the number of classes, aligning closely with the diagonal representing perfect correlation. PLDiv, however, offers faster computation and slightly higher correlation. DCScore follows, showing comparable performance with one embedding model but greater variance with the other. In contrast, Vendi Score tends to decrease as the number of classes and data increases. This indicates that the geometry-aware property of PLDiv makes it particularly well-suited for vision tasks, where embeddings often encode the geometric structure of images.

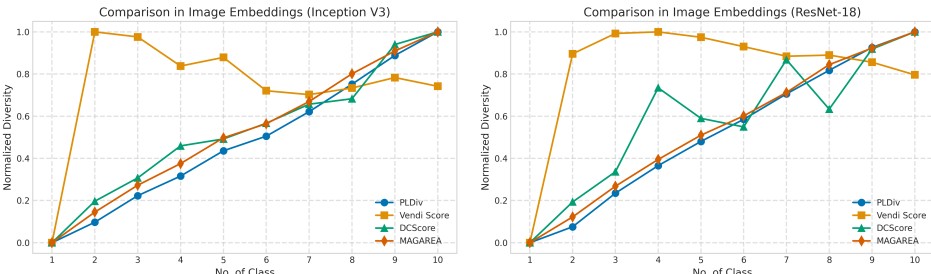

Figure 5: PLDiv shows a near-perfect correlation with the amount of the class involved in the dataset and remains consistent across different embedding models. MAGAREA performs next best, followed by DCScore, which exhibits some fluctuations in performance. VS, however, fails to capture the underlying patterns in the data.

## 5.5 DIVERSITY ASSESSMENT IN SYNTHETIC DATA CLOUDS

To demonstrate that PLDiv serves as a geometry-aware diversity metric, we simulated eight pairs of two-dimensional point clouds (A, B), each containing about 200 points generated from parameterized geometric functions described in Appendix Table 7. Each pair modifies one specific geometric property by adding or removing loops, bridges, curvature, or hierarchical clustering, while maintaining a comparable overall spatial scale. These controlled scenarios allow a direct comparison of how different metrics respond to structural variation rather than random dispersion.

We computed PLDiv, Vendi Score, DCScore, and MagArea on Euclidean distance matrices for each dataset. A metric is considered consistent if it assigns a higher diversity value to the configuration exhibiting richer geometric organization. PLDiv meets this criterion across all eight cases, while Vendi Score and MagArea do so in seven and DCScore in only three. Moreover, PLDiv produces sharper and directionally coherent contrasts between paired clouds. For instance, *Ring vs Disk* and *Nested vs Gaussian* exhibit strong PLDiv separation that quantitatively reflects the presence or loss of loops, whereas the other metrics change only slightly. The difference arises from what each measure encodes: Vendi Score and DCScore emphasize global similarity spectra or density separation, and MagArea summarizes scale magnitude but not connectivity. PLDiv, by integrating the persistence of topological features across filtrations, captures differences that are geometrically meaningful and also visually intuitive, as illustrated in Fig. 6.

## 5.6 COMPUTATION COMPLEXITY

In this section, we analyze the computational cost of our proposed metric compared with existing approaches. When the input is a point cloud $\mathcal{X} \in \mathbb{R}^{n \times d}$, computing all pairwise distances requires $\mathcal{O}(n^2 d)$ time, whereas utilizing a precomputed distance matrix sets the baseline at $\mathcal{O}(n^2)$. While standard persistent homology and PLDiv computation scale quadratically with $n$ due to the number of edges, their effective cost can be substantially reduced via sparsification. Specifically, the sparse Rips filtration (Cavanna et al., 2015) utilizes a tolerance parameter $\epsilon$ to construct a $(1 + \epsilon)$-approximation of the metric space. This method prunes the graph to a linear size $\mathcal{O}(C(\epsilon)n)$; since $C(\epsilon)$ scales inversely with $\epsilon$, larger tolerances significantly accelerate computation with negligible accuracy loss (see Table 3). We then compute PLDiv using the Minimum Spanning Tree (MST) of the sparse Rips graph, a strategy that reduces the standard $\mathcal{O}(n^2)$ time and memory complexity of dense methods to near-linear time and linear $\mathcal{O}(n)$ memory. Finally, PLDiv can be computed via a closed-form expression in $\mathcal{O}(N_d)$ time, outperforming the Vendi Score on large-scale benchmarks (see Table 2).

## 6 CONCLUSION

Understanding data diversity requires moving beyond traditional notions of variation or entropy to account for the intricate geometric and topological structures inherent in complex datasets. We propose a geometry-aware data diversity measure based on persistence landscapes, a tool from topolog-

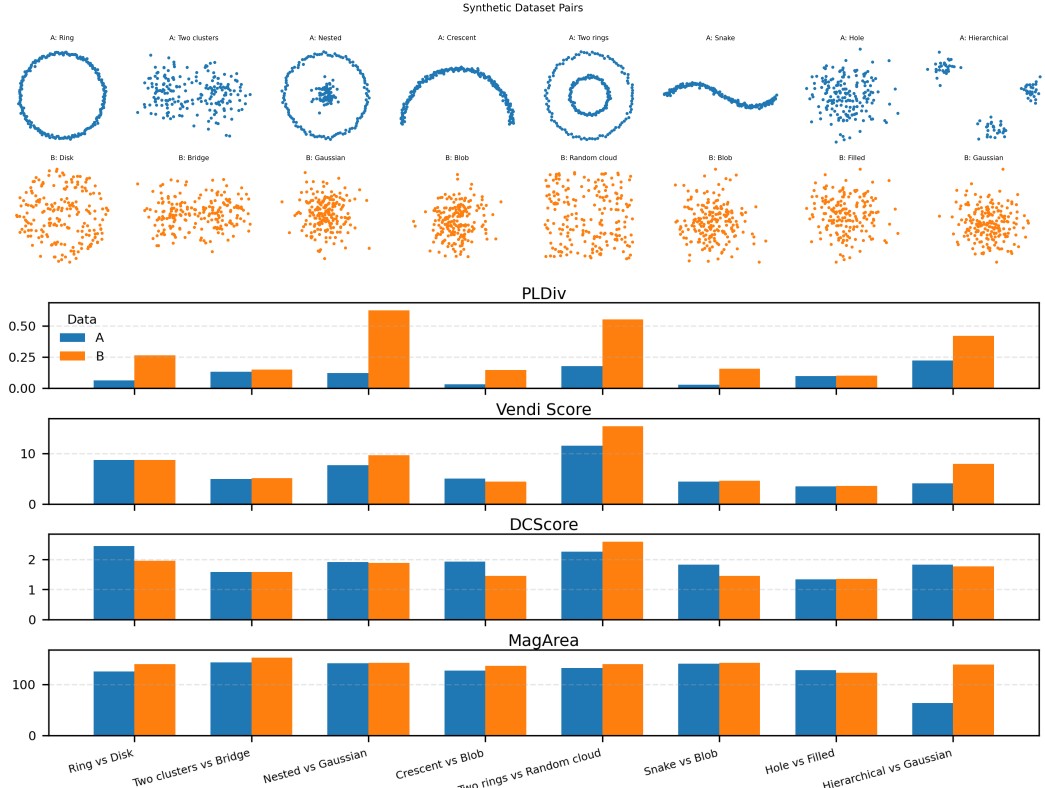

Figure 6: Synthetic dataset comparison. *Upper:* eight dataset pairs (A vs. B), each with 200 points, generated to introduce or remove loops, bridges, or hierarchical clusters. *Lower:* diversity scores across metrics. PLDiv yields sharper and more coherent distinctions that reflect the true geometric differences between datasets, while Vendi Score, DCScore, and MagArea respond mainly to overall spread and fail to capture these structural changes in most cases.

Table 2: Computation time comparison with varying sample sizes on ImageNet-1K. Embeddings are extracted using ResNet-50 and computed based on cosine similarity/distance. Values are reported in seconds.

| Method | Sample size (ImageNet-1K) | | | | |
|---|---|---|---|---|---|
| | 5k | 10k | 20k | 30k | 40k |
| Vendi Score | $1.60_{\pm 0.83}$ | $10.82_{\pm 2.73}$ | $183.80_{\pm 12.88}$ | $746.51_{\pm 30.74}$ | $1786.11_{\pm 184.64}$ |
| DCScore | $0.03_{\pm 0.02}$ | $0.13_{\pm 0.01}$ | $0.46_{\pm 0.01}$ | $1.00_{\pm 0.01}$ | $1.80_{\pm 0.05}$ |
| MAGAREA | $164.91_{\pm 29.55}$ | $716.14_{\pm 31.23}$ | – | – | – |
| **PLDiv** | $5.43_{\pm 0.02}$ | $24.33_{\pm 0.09}$ | $105.62_{\pm 0.35}$ | $236.23_{\pm 0.76}$ | $462.75_{\pm 0.56}$ |
| **Sparse PLDiv** ($\epsilon = 0.95$) | $3.97_{\pm 0.03}$ | $16.80_{\pm 0.37}$ | $68.55_{\pm 2.21}$ | $147.48_{\pm 6.50}$ | $273.86_{\pm 14.35}$ |
| **Sparse PLDiv** ($\epsilon = 10$) | $2.61_{\pm 0.00}$ | $9.87_{\pm 0.05}$ | $33.74_{\pm 0.01}$ | $68.15_{\pm 0.76}$ | $115.54_{\pm 0.24}$ |

Table 3: Sparse PLDiv values demonstrating its reliable computation

| Method | Sample size | | | | |
|---|---|---|---|---|---|
| | 5k | 10k | 20k | 30k | 40k |
| **PLDiv** | 46.51 | 78.01 | 133.55 | 184.93 | 232.89 |
| **Sp. PLDiv** ($\epsilon = 0.95$) | 46.52 | 78.03 | 133.58 | 184.92 | 232.89 |
| **Sp. PLDiv** ($\epsilon = 10$) | 47.32 | 79.70 | 136.86 | 190.23 | 240.04 |

ical data analysis that provides a stable and expressive representation of hidden structural patterns. Our metric, PLDiv, offers a richer and more nuanced quantification of diversity. Through extensive experiments across multiple domains and modalities, we demonstrate PLDiv's ability to characterize structural properties in data clouds (e.g., curvature data) and in vector embeddings (e.g., text and image data). These results suggest that PLDiv provides a principled foundation for analyzing geometric diversity, with potential applications in dataset construction, augmentation, model evaluation, and robustness analysis. Looking forward, integrating topological perspectives into automated dataset design, generative modeling, and adaptive learning systems has the potential to fundamentally reshape how diversity is understood, measured, and leveraged in artificial intelligence.

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

## A    ADDITIONAL ITERATURE REVIEW

### A.1    DIVERSITY MEASUREMENT

Evaluating diversity has long been a challenge in machine learning and generative modeling, partly because it is not always formalized under a single definition but manifests across different dimensions. For example, holistic evaluations of language models highlight variation in task coverage, domain shifts, linguistic and dialectal richness, input perturbations, and social context, all of which directly connect to the broader notion of data diversity (Liang et al., 2022).

Some works emphasize that inducing or controlling diversity can be as important as measuring it. Behavioral frameworks such as CheckList (Ribeiro et al., 2020) systematically probe models through templating, lexical substitutions, and perturbations, showing that diverse inputs are essential for revealing hidden model failures, even though diversity itself is not explicitly quantified.

Diversity is not always treated only as an evaluation objective, but also as a design principle at the training level. For instance, Du and Black (Du & Black, 2019) mitigate mode collapse in dialogue generation by iteratively boosting models to promote semantic and lexical variation. Although effective in practice, these approaches underscore the need for principled evaluation frameworks that can verify whether training-time interventions truly enhance diversity across settings.

To address semantic variation more directly, semantic diversity methods examine conceptual distinctions between outputs. Stasaski and Hearst (Stasaski & Hearst, 2022) use Natural Language Inference models to identify entailment, contradiction, and neutrality among generated texts, treating contradiction as a marker of diversity and entailment as redundancy. Although intuitive and fine-grained, this relational approach is inherently limited to pairwise comparisons and does not capture global structural diversity across datasets.

A large class of methods focuses on surface-level variation, particularly in text. N-gram–based metrics such as distinct-n (Song et al., 2024), self-BLEU (Shu et al., 2019), and ROUGE-L (Wang et al., 2022; Padmakumar & He, 2023) capture token-level dispersion across samples (Yu et al.,

2017). Similarly, the Data Quality Index (DQI) (Mishra et al., 2020) aggregates vocabulary richness, entropy, and syntactic variation to assess dataset quality. While easy to compute, these approaches provide only a narrow view of diversity, often missing deeper semantic or structural patterns.

## A.2 Persistent Homology in Metric Space

The formal algebraic foundations were established by Zomorodian & Carlsson (2004), who introduced persistence modules, provided algorithms for computing persistence, and proved the barcode decomposition theorem as a complete invariant over fields. This work grounded PH in computability and algebraic classification, laying the basis for its adoption across domains (Zhao & Wang, 2019; Hiraoka et al., 2016; Pun et al., 2022). However, these foundational contributions primarily emphasize topology extraction and stability, without directly connecting persistence to data-level diversity or representational richness.

Beyond its theoretical foundations, TDA and persistent homology have shown practical utility across diverse domains. In neuroscience, PH captures vascular structures linked to disease (Bendich et al., 2016); in materials science, it characterizes microstructures and force chains in amorphous solids (Hiraoka et al., 2016); and in biology and chemistry, it reveals topological signatures of protein folding, molecular stability, and binding sites (Xia & Wei, 2015; Kovacev-Nikolic et al., 2016; Gameiro et al., 2015). These examples highlight PH's ability to extract robust, multi-scale features from high-dimensional and noisy data.

PH has also been applied to both temporal and spatial systems. Persistence landscapes have been used to track transitions in dynamical systems and classify time-series data (Gidea & Katz, 2018; Umeda, 2017), while in astrophysics, PH captures the multiscale filamentary structure of the cosmic web from cosmological simulations (Aragón-Calvo et al., 2010). Collectively, these applications highlight PH's versatility as a modality-agnostic framework for extracting global, nonlinear structure that often remains inaccessible to conventional statistical or machine learning methods.

# B DESCRIPTION OF DIVERSITY SCORES IN COMPARISONS

Vendi Score (VS) (Dan Friedman & Dieng, 2023), derived from a set of samples and their pairwise similarity functions, quantifies the similarities among the data in a dataset. Mathematically, VS is given by the exponential of the Shannon entropy, which is obtained from the eigenvalues of the scaled similarity matrix $X^\top X$:

$$VS = \exp\left(-\sum_{i=1}^{n} \lambda_i \log \lambda_i\right)$$

where $\lambda_i$ are the eigenvalues of scaled $X^\top X$.

Limbeck et al. (2024) introduces several *magnitude-based* diversity measures that leverage the notion of the effective size of a metric space across scales. The core idea is to compute the *magnitude function*, $\mathrm{Mag}_X(t)$, which tracks how the effective number of points in a space changes as pairwise distances are rescaled. To summarise this behaviour, the authors propose two derived metrics: the area under the magnitude function (MAGAREA) as a reference-free measure of intrinsic diversity, and the difference between magnitude functions (MAGDIFF) as a reference-based measure:

$$\mathrm{MAGAREA} = \int_{t_0}^{t_{\mathrm{cut}}} \mathrm{Mag}_X(t)\, dt, \quad \mathrm{MAGDIFF} = \int_{t_0}^{t_{\mathrm{cut}}} \left(\mathrm{Mag}_X(t) - \mathrm{Mag}_Y(t)\right) dt,$$

where $\mathrm{Mag}_X(t)$ is the magnitude function of $X$ at scale $t$ and $t_{\mathrm{cut}}$ denotes the convergence scale used for evaluation. These measures provide robust multi-scale summaries of diversity and have been shown to detect phenomena such as curvature, mode collapse, and mode dropping in text, image, and graph representations.

Zhu et al. (2025) proposes **DCScore**, which departs from entropy or scale-based approaches by reframing diversity measurement as a *classification problem*. Instead of relying on eigenvalue decomposition or scale-sensitive geometric measures, DCScore evaluates how well each individual sample in a dataset can be distinguished from all others. Specifically, each sample is treated as its

own class, and pairwise similarities are converted into classification probabilities through a softmax function. The last score is then defined as the trace of the resulting probability matrix:

$$\text{DCScore}(D) = \text{tr}(P) = \sum_{i=1}^{n} P[i, i], \quad P[i, j] = \frac{\exp\left(\frac{K[i,j]}{\tau}\right)}{\sum_{k=1}^{n} \exp\left(\frac{K[i,k]}{\tau}\right)},$$

where $K[i, j]$ denotes the similarity between samples $i$ and $j$, and $\tau$ is a temperature parameter that controls the classification sharpness. This formulation is principled and efficient, emphasizing sample separability without considering the geometric or topological structure of the dataset, which can also be important for characterizing diversity.

## C  MATHEMATICAL PROOFS

### C.1  PLDIV CLOSED FORM

Let $\mathcal{D} = \{(b_i, d_i)\}_{i=1}^{m}$ be a finite multiset of persistence birth–death pairs and let $\lambda_{(b_i, d_i)} : \mathbb{R} \to [0, \infty)$ denote the usual persistence "tent" function associated to the interval $(b_i, d_i)$. Let $\{\lambda_k(t)\}_{k \geq 1}$ be the persistence landscape functions obtained by ordering the values $\{\lambda_{(b_i, d_i)}(t)\}_{i=1}^{m}$ at each fixed $t$ in nonincreasing order (with $\lambda_k(t) = 0$ for all $k > m$). Then

$$\text{PLDiv}(\mathcal{X}) = \sum_{k=1}^{\infty} \int_{\mathbb{R}} \lambda_k(t)\, dt = \sum_{i=1}^{m} \int_{\mathbb{R}} \lambda_{(b_i, d_i)}(t)\, dt = \frac{1}{4} \sum_{i=1}^{m} (d_i - b_i)^2.$$

*Proof.* By definition $\lambda_k(t)$ are the order statistics (at each fixed $t$) of the family $\{\lambda_{(b_i, d_i)}(t)\}_{i=1}^{m}$. For any finite collection of nonnegative functions $f_i(t)$,

$$\sum_{k=1}^{\infty} k\text{-th largest of } \{f_i(t)\} = \sum_{i=1}^{m} f_i(t),$$

Applying this pointwise gives

$$\sum_{k=1}^{\infty} \lambda_k(t) = \sum_{i=1}^{m} \lambda_{(b_i, d_i)}(t).$$

Each $\lambda_{(b_i, d_i)}$ is continuous with compact support $[b_i, d_i]$, hence measurable and integrable. By Tonelli's theorem (Tao, 2011),

$$\sum_{k=1}^{\infty} \int_{\mathbb{R}} \lambda_k(t)\, dt = \int_{\mathbb{R}} \sum_{k=1}^{\infty} \lambda_k(t)\, dt = \int_{\mathbb{R}} \sum_{i=1}^{m} \lambda_{(b_i, d_i)}(t)\, dt = \sum_{i=1}^{m} \int_{\mathbb{R}} \lambda_{(b_i, d_i)}(t)\, dt.$$

Finally, each tent function supported on the interval $[b_i, d_i]$ is a symmetric isosceles triangle of base length $d_i - b_i$ and height $(d_i - b_i)/2$, hence its area is

$$\int_{\mathbb{R}} \lambda_{(b_i, d_i)}(t)\, dt = \tfrac{1}{2} \cdot (d_i - b_i) \cdot \tfrac{d_i - b_i}{2} = \frac{(d_i - b_i)^2}{4},$$

Summing over $i = 1, \dots, m$ gives the final identity

$$\sum_{i=1}^{m} \int_{\mathbb{R}} \lambda_{(b_i, d_i)}(t)\, dt = \frac{1}{4} \sum_{i=1}^{m} (d_i - b_i)^2.$$

$\square$

## C.2 PROOF OF AXIOMATIC PROPERTIES OF DIVERSITY

A diversity measure derived from Persistence Landscapes (PLs) is defined as a summary statistic of the persistence lifetimes generated from a dataset's Vietoris-Rips filtration. We prove that such a measure satisfies the key principles of effective size, monotonicity, the twin property, and symmetry.

- **Effective size.** For a fixed number of points, $\mathrm{PLDiv}(\mathcal{X})$ increases when data points are well-separated and decreases as they cluster, reaching a maximum when all points are distinct and a minimum when all are identical.

  *Proof. Minimum* PLDiv*:* The minimum value of PLDiv is achieved when all points in the cloud $\mathcal{X}$ are identical. Let all $n$ points in the cloud be the same, so $x_1 = x_2 = \cdots = x_n$. The distance between any two points is zero:

  $$d(x_i, x_j) = 0 \quad \text{for all } i, j.$$

  Every point is born at $\varepsilon = 0$ and immediately merges with every other point at $\varepsilon = 0$, all persistence lifetimes are zero. That is,

  $$b_i = 0, \quad d_i = 0 \quad \text{for all features.}$$

  Therefore,

  $$\min \mathrm{PLDiv}(\mathcal{X}) = \frac{1}{4} \sum_i (d_i - b_i)^2 = \frac{1}{4} \sum_i (0 - 0)^2 = 0.$$

  *Maximum* PLDiv*:* The maximum value of PLDiv is achieved when the points are "well-separated." Let $\mathcal{X} = \{x_1, \ldots, x_n\}$ be a point cloud in a metric space $(\mathcal{M}, d)$ such that all points are distinct and equidistant:

  $$d(x_i, x_j) = c > 0 \quad \text{for all } i \neq j.$$

  Then, the $H_0$ persistence lifetimes are all equal to $c$, except for the last surviving component. Let $c = \max_{i \neq j} d(x_i, x_j)$. In the Vietoris–Rips filtration, at $\varepsilon = 0$, each point forms a separate connected component. Thus, there are $n$ components born at $b_i = 0$. For $0 < \varepsilon < c$, no edges appear because all pairwise distances are $c$. Hence, no components merge in this interval. At $\varepsilon = c$, all pairwise edges appear simultaneously, and the $n$ components merge into a single connected component. Thus, $n - 1$ components die at $d_i = c$, while the last component persists indefinitely.

  By Proposition 3.2, the corresponding PLDiv is

  $$\max \mathrm{PLDiv}(\mathcal{X}) = \frac{n-1}{4} c^2.$$

  $\square$

- **Monotonicity**

  Fix $n$ and let $\mathcal{X}$ be a point cloud in a metric space. If all pairwise distances in $\mathcal{X}$ are scaled by a factor $\alpha > 1$ (i.e. replace the metric $d(\cdot, \cdot)$ by $\alpha d(\cdot, \cdot)$), then

  $$\mathrm{PLDiv}(\alpha \mathcal{X}) \begin{cases} \leq \alpha^2 \, \mathrm{PLDiv}(\mathcal{X}), & \alpha > 1, \\ \geq \alpha^2 \, \mathrm{PLDiv}(\mathcal{X}), & 0 < \alpha < 1. \end{cases}$$

  *Proof.* Fix $n$ and let $\mathcal{X}$ be a point cloud in a metric space. If all pairwise distances in $\mathcal{X}$ are scaled by a factor $\alpha > 1$ (i.e. replace the metric $d(\cdot, \cdot)$ by $\alpha d(\cdot, \cdot)$), then every lifetime $d_i - b_i$ is multiplied by $\alpha$. By Proposition 3.2,

  $$\mathrm{PLDiv}(\alpha \mathcal{X}) = \frac{1}{4} \sum_i (\alpha(d_i - b_i))^2 = \alpha^2 \cdot \frac{1}{4} \sum_i (d_i - b_i)^2 = \alpha^2 \mathrm{PLDiv}(\mathcal{X}).$$

  Hence, spreading the same set of points apart (uniform dilation) strictly increases PLDiv (for $\alpha > 1$). More generally, moving points so as to increase lifetimes of the dominant features increases PLDiv; conversely, clustering points tends to shorten lifetimes and reduce PLDiv. $\square$

- **Twin property.** Adding an exact duplicate of a point does not change $\text{PLDiv}(\mathcal{X})$. Let $\mathcal{X}$ be a dataset and let $x_i \in \mathcal{X}$. For the set $\mathcal{X}' = \mathcal{X} \cup \{x_n\}$ where $x_n = x_i$, the diversity is unchanged:

$$\text{PLDiv}(\mathcal{X}') = \text{PLDiv}(\mathcal{X}).$$

  *Proof.* A duplicate point at exactly the same coordinates is at zero distance from its twin. In the usual filtrations built from pairwise distances (e.g., Vietoris–Rips), the duplicate component is born at radius $0$ and immediately merges with its twin also at radius $0$. Hence the corresponding birth–death pair is $(0, 0)$ and has lifetime $0$, contributing $(d - b)^2/4 = 0$ to the PLDiv sum. All other birth–death pairs are unchanged as well. Therefore PLDiv is unchanged. $\square$

- **Symmetry.** PLDiv is invariant to the ordering of data points (permutation invariance). Since persistent homology depends only on the metric structure of $\mathcal{X}$ and $\text{PLDiv}(\mathcal{X})$ is computed from the multiset of intervals $\{(b_i, d_i)\}$, relabeling or reordering points does not affect the value of the score. Let $\mathcal{X} = (x_1, \ldots, x_n)$ be an ordered sequence of points and let $\pi$ be any permutation of $\{1, \ldots, n\}$. For the permuted sequence $\mathcal{X}_\pi = (x_{\pi(1)}, \ldots, x_{\pi(n)})$, we have

$$\text{PLDiv}(\mathcal{X}_\pi) = \text{PLDiv}(\mathcal{X}).$$

  *Proof.* The PH pipeline begins with the pairwise distance matrix $D$, where $D_{ij} = d(x_i, x_j)$. Let $\mathcal{X}_\pi$ be the reordered dataset. The distance matrix $D_\pi$ for the permuted data has entries $(D_\pi)_{ij} = d(x_{\pi(i)}, x_{\pi(j)})$. Importantly, the set of all unique pairwise distances

$$\{d(x_i, x_j)\}_{1 \le i < j \le n}$$

  is unchanged for both $\mathcal{X}$ and $\mathcal{X}_\pi$. The construction of the Vietoris–Rips filtration depends only on these distances. Hence, the persistence diagrams and lifetimes $\{l_i\}$ are identical. Therefore, any diversity measure computed from these lifetimes is invariant under permutation of the data and PLDiv is symmetry. $\square$

# D  DETAILED EXPERIMENT DESCRIPTIONS

## D.1  SYNTHETIC TOY EXAMPLES

Our toy example in Figure 1 utilizes the examples from Limbeck et al. (2024). Specifically, we simulated four synthetic datasets with varying diversity levels. D1 (Poisson Process): 200 points uniformly sampled in the square $[0, 2]^2$, representing a spatially random distribution. D2 (Hawkes Process): a clustered dataset generated via a self-exciting point process with base intensity $\lambda = 91$ and excitation parameter $\alpha = 0.6$. D3 (Two Gaussians): 200 samples drawn from two Gaussian clusters centered at $(0.5, 0.5)$ and $(1.5, 1.5)$ with covariance $0.02I$. D4 (One Gaussian): 200 samples drawn from a single Gaussian centered at $(0.5, 0.5)$ with the same covariance. These datasets progressively transition from highly diverse and dispersed (D1) to concentrated and homogeneous (D4). Table 4 represents diversity scores calculated by four metrics. (Vendi Score and DCScore are based on RBF kernel)

Table 4: Performance comparison of subset selection

| Task | PLDiv ($\uparrow$) | Vendi Score (rbf) ($\uparrow$) | DCScore ($\uparrow$) | MagArea ($\uparrow$) |
|------|------|------|------|------|
| D1 | 0.53 | 136.98 | 2.67 | 141.23 |
| D2 | 0.49 | 79.96 | 2.63 | 108.83 |
| D3 | 0.26 | 40.40 | 2.48 | 81.93 |
| D4 | 0.05 | 23.66 | 2.32 | 58.53 |

### D.1.1 IMBALANCED SYNTHETIC DATA

To explore how PLDiv performs on imbalanced data, we generated a series of small long-tail datasets. First, we utilized D4 in synthetic toy examples, which form a single cluster with 200 data points. To simulate long-tail effects, outlier points were added uniformly within a square region in varying amounts of 20, 40, 60, 80, and 100 samples, while keeping the cluster size at 200 - $n_{\text{outliers}}$. Each variant thus exhibits increasing imbalance between the dense Gaussian core and sparse tail regions. Figure 7 demonstrates that PLDiv effectively handles the imbalanced dataset.

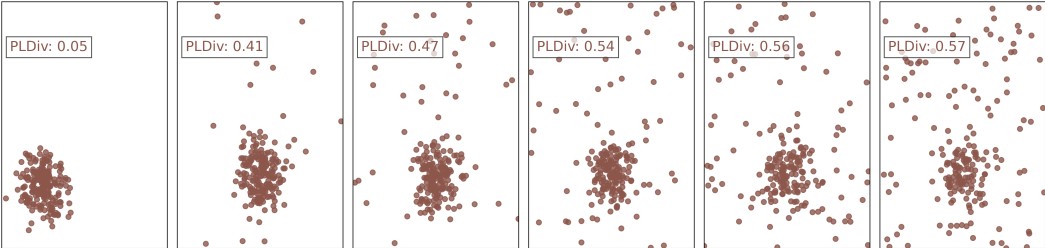

Figure 7: PLDiv can reliably predict diversity in imbalanced data, where diversity increases monotonically.

### D.2 IMPLEMENTATION OF $k$-DPP SAMPLING

To select a diverse subset of $k$ instances, we implemented a $k$-Determinantal Point Process ($k$-DPP) (Kulesza & Taskar, 2011). Two Gaussian clusters were generated and combined to form the dataset. Each cluster consisted of 100 points drawn from a Gaussian distribution with means 0.5 and 0.6, and a standard deviation of 0.05. An RBF kernel was computed using the median pairwise distance as the bandwidth parameter:

$$k_{ij} = \exp\left(-\frac{\|x_i - x_j\|^2}{2\sigma^2}\right)$$

The kernel matrix was eigendecomposed, and the top-$k$ eigenvectors corresponding to the largest eigenvalues were retained. Points were then iteratively sampled with probabilities proportional to the squared norms of these eigenvectors. After each selection, the eigenbasis was orthogonalized to maintain diversity. This procedure yielded $k$ representative and diverse samples from the original dataset. Similarly, we applied the same approach to the ArXiv dataset to create the k-DPP subset. Table 5 presents the results of the diversity measures, illustrating how they capture the subtle differences between random selection and k-DPP selection.

Table 5: Performance comparison of subset selection

| Task | PLDiv | Vendi Score | DCScore | MagArea |
|------|-------|-------------|---------|---------|
| simulation (random) | 0.009 | 1.051 | 1.007 | 19.645 |
| simulation (KDPP) | 0.018 | 1.099 | 1.016 | 23.340 |
| ArXiv (random) | 25.392 | 39.729 | 2.132 | 40.507 |
| ArXiv (KDPP) | 26.620 | 43.175 | 2.185 | 41.422 |

### D.3 IMPLEMENTATION OF CURVATURE EXPERIMENT

In Section 5.2, we evaluate PLDiv along with alternative diversity metrics on the curvature dataset (Turkes et al., 2022). The dataset consists of two-dimensional point clouds sampled from smooth surfaces with varying degrees of curvature. Each sample represents a set of points $\{x_i\}_{i=1}^n \subset \mathbb{R}^d$ labeled by the curvature of the underlying manifold, either as discrete curvature classes or continuous curvature values, ranging from -2 to 2. The task is to predict this curvature from the sampled points, assessing how well diversity measures capture geometric information such as local bending and shape variation. This setup allows controlled evaluation of geometric sensitivity, robustness to noise, and invariance under isometric transformations.

We employ a Support Vector Regression (SVR) model with a radial basis function (RBF) kernel, using the parameters C = 1.0 and $\epsilon = 0.1$. This configuration is applied to all metrics (PLDiv, Vendi Score, DCScore, and MagArea). MagArea uses Euclidean distance, while Vendi Score and DCScore are evaluated with both RBF and Laplacian kernels. In contrast, PLDiv takes the curvature data cloud as input and internally computes pairwise Euclidean distances. Table 1 and Figure 8 demonstrate that PLDiv exhibits a truly geometry-aware property.

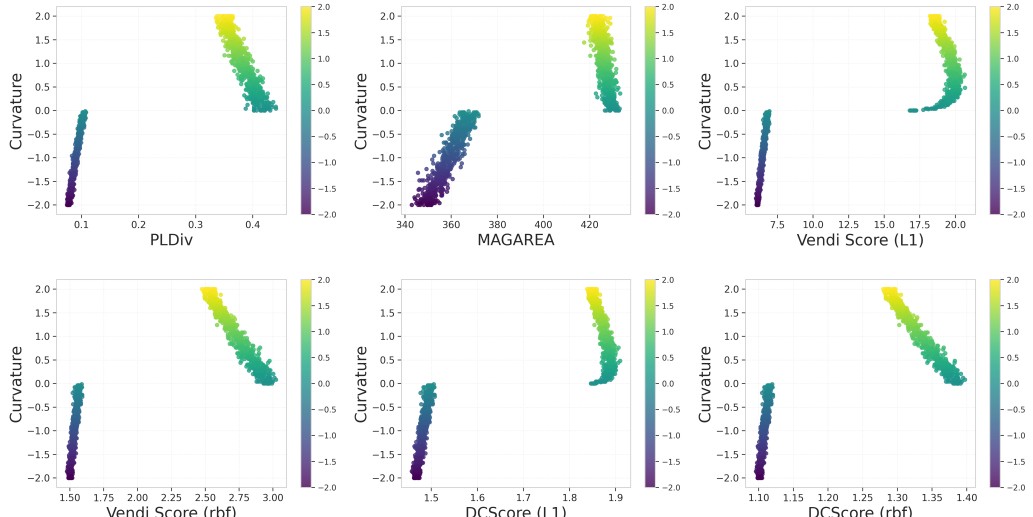

Figure 8: Visualizations of the diversity measures against the curvature labels show that PLDiv achieves the best separation between positive and negative curvatures, providing clear evidence of why it performs best in Section 5.2.

### D.4 IMPLEMENTATION OF TEXT EMBEDDINGS

We evaluate PLDiv as a metric of semantic diversity using the dataset from Tevet & Berant (2021), comprising 1,000 prompts from three tasks. Ten outputs per prompt were generated by varying the softmax temperature (*dec*), and a subset of 200 prompts was human-annotated to obtain mean diversity scores (*ABS-HDS*). Text embedding models we used are listed below:

- all-MiniLM-L12-v2: general text embedding model, dimension 384
- all-mpnet-base-v2: general text embedding model, dimension 768
- bert-large-nli-stsb-mean-tokens: general text embedding model, dimension 1024
- Qwen3-Embedding-4B: advanced LLM-based embedding models, dimension 2560
- Qwen3-Embedding-8B: advanced LLM-based embedding models, dimension 4096

Figure 9 represents Mean Squared Error (MSE) for linear regression that indicates the predictive capability for diversity metrics on softmax temperature *dec* and mean human annotated diversity score (*ABD-HDS*). PLDiv achieves the lowest MSE in the temperature (*dec*) tasks across all embedding models and remains among the lowest when evaluated on human-annotated scores.

To explore the impact of the distance/similarity matrix, we applied both cosine distance/similarity and Euclidean distance/RBF kernel as inputs in this experiment ihe temperature (dec) tasks. Figure 10 demonstrates that PLDiv consistently and reliably outperforms other metrics across various embedding models and distance matrices. In contrast, switching from cosine similarity to the RBF kernel significantly degrades the performance of alternative metrics, particularly DCScore.

We present the correlation plots for text embedding temperature *dec* evaluation tasks in Figs. 11, 12, and 13. Across the three embedding tasks, PLDiv shows the best performance on all three tasks: prompt, response, and story, exhibiting a linear relationship, while providing a non-linear relationship with softmax temperature *dec* .

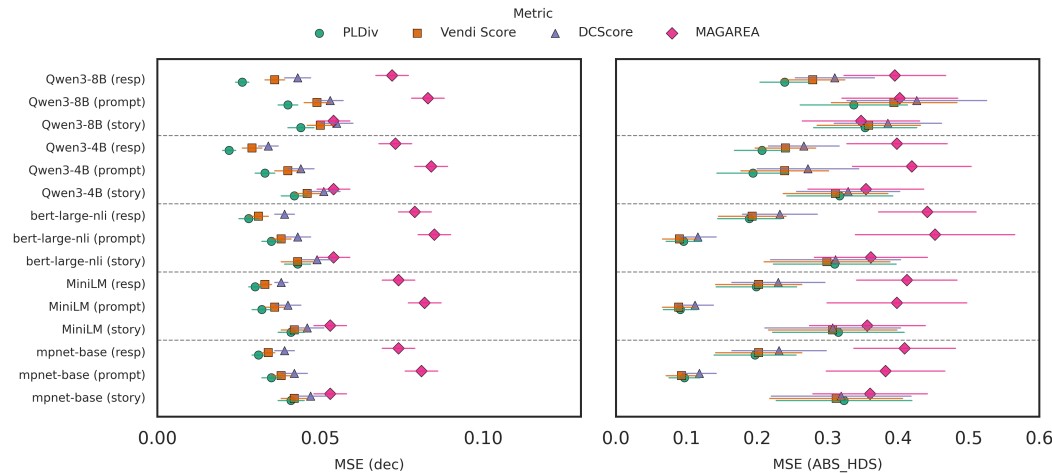

Figure 9: MSE for four metrics on both temperature *dec* and human diversity score *ABD-HDS*. PLDiv achieves the lowest MSE in the temperature (*dec*) tasks across all embedding models and remains among the lowest when evaluated on human-annotated scores.

### D.5 IMPLEMENTATION OF IMAGE EMBEDDINGS

In Section 5.4, we evaluated the diversity measure to determine whether it can effectively capture the diversity introduced by the richness of labels. We employed Colored MNIST Deng (2012). Following the methodology of Ospanov et al. (2024), the number of labels served as the ground truth for diversity, where a higher label count signifies a more diverse set. We sampled half of the data from each class. Starting from class 1, we incrementally added samples from one additional class at a time, up to class 10, thereby forming 10 subsets. Comparisons are conducted against Vendi Score, Magnitude, and DCScore, using two embedding models: Inception V3 and ResNet-18. All metrics are tested on cosine distance or cosine similarity. Figure 5 and Table 6 show that PLDiv can effectively capture diversity encoded in image embeddings. PLDiv achieved comparable results with MagArea but is more computationally efficient.

Table 6: Pearson Correlation Comparison among diversity measures

| Metric | CLIP Model | Inception Model |
|---|---|---|
| PLDiv | 0.998 | 0.998 |
| Vendi Score | 0.371 | 0.222 |
| DCScore | 0.901 | 0.984 |
| MagArea | 0.997 | 0.998 |

### D.6 DIVERSITY ASSESSMENT IN SYNTHETIC DATA CLOUDS DETAILS

We created eight pairs of synthetic scenarios, each containing about 200 points generated from parameterized geometric functions. Each pair modifies one specific geometric property by adding or removing loops, bridges, curvature, or hierarchical clustering, while maintaining a comparable overall spatial scale. Table 7 summarizes the data generation process for the eight synthetic point-cloud pairs used in Sec. 5.4. Each cloud contains approximately 200 points produced by explicit geometric or probabilistic functions (e.g., rings, Gaussian mixtures, sinusoidal manifolds). These datasets complement Table 8, which reports diversity metric values across the same scenarios.

## E LIMITATIONS

While PLDiv demonstrates strong theoretical grounding and robust empirical performance across modalities, we acknowledge several limitations and areas for future improvement. First, computational cost is not the primary focus of this work. Although we proposed a sparse computation

Table 7: Synthetic dataset pairs used for geometry-aware diversity evaluation. Each cloud contains 200 points.

| Pair | A (less varied geometry) | B (more varied geometry) |
|---|---|---|
| Ring vs Disk | Uniform points in filled disk | Points on noisy circular rim (loop) |
| Two Clusters vs Bridge | Two separated Gaussian blobs | Same blobs plus short bridge (connectivity) |
| Gaussian vs Nested | Single Gaussian | Inner Gaussian + outer ring (hierarchy) |
| Blob vs Crescent | Isotropic Gaussian cloud | Half-ring manifold (curvature) |
| Random Cloud vs Two Rings | Uniform on square $[0, 2]^2$ | Two concentric noisy rings (multi-loop) |
| Blob vs Snake | Isotropic Gaussian | Sinusoidal curve with noise (manifold) |
| Filled vs Hole | Outer Gaussian + center points | Outer Gaussian with inner void (cavity) |
| Gaussian vs Hierarchical | Single broad Gaussian | Multi-level small clusters (multi-scale) |

Table 8: Comparison of diversity metrics across synthetic dataset pairs.

| Scenario | Data | PLDiv | Vendi Score | DCScore | MagArea |
|---|---|---|---|---|---|
| Ring vs Disk | A | 0.064 | 8.702 | 2.437 | 125.732 |
| Ring vs Disk | B | 0.262 | 8.746 | 1.957 | 140.620 |
| Two clusters vs Bridge | A | 0.134 | 4.915 | 1.578 | 143.599 |
| Two clusters vs Bridge | B | 0.150 | 5.132 | 1.585 | 153.364 |
| Nested vs Gaussian | A | 0.123 | 7.696 | 1.906 | 141.750 |
| Nested vs Gaussian | B | 0.623 | 9.641 | 1.878 | 142.509 |
| Crescent vs Blob | A | 0.030 | 5.025 | 1.919 | 127.702 |
| Crescent vs Blob | B | 0.147 | 4.469 | 1.450 | 136.976 |
| Two rings vs Random cloud | A | 0.176 | 11.569 | 2.257 | 132.447 |
| Two rings vs Random cloud | B | 0.551 | 15.436 | 2.583 | 140.364 |
| Snake vs Blob | A | 0.027 | 4.405 | 1.827 | 141.067 |
| Snake vs Blob | B | 0.156 | 4.589 | 1.455 | 142.696 |
| Hole vs Filled | A | 0.096 | 3.458 | 1.342 | 128.140 |
| Hole vs Filled | B | 0.101 | 3.559 | 1.352 | 122.926 |
| Hierarchical vs Gaussian | A | 0.222 | 4.048 | 1.824 | 63.258 |
| Hierarchical vs Gaussian | B | 0.420 | 7.972 | 1.768 | 139.101 |

that significantly reduces both time and memory requirements, PLDiv remains computationally intensive than lightweight alternatives such as DCScore. Our contribution emphasizes accuracy and geometric faithfulness rather than speed, and we recognize that there is room for further algorithmic optimization.

Second, PLDiv currently employs the Vietoris–Rips filtration as its default topological construction. While this choice offers broad applicability and simplicity, alternative filtrations, such as Čech, Alpha Complex, etc, may capture structure more effectively in specific domains. Exploring these variants could further increase the flexibility of PLDiv.

Third, PLDiv balances fine-grained local feature capture with preservation of global geometric structure, governed by the maximum-edge parameter. In our experiments, a single global setting was sufficient, though in other specific cases, this parameter may need tuning to balance local sensitivity and computational efficiency.

## F    COMPUTATIONAL ENVIRONMENT

All experiments were conducted on a high-performance computing server equipped with an AMD EPYC 7413 24-Core Processor and an NVIDIA A100-80GB GPU. The software environment was built using Python 3.11. For text embedding, we utilized Hugging Face Sentence Transformers as the embedding model framework.

Table 9: Additional Computation time comparison. (the value scale in seconds)

| Method | Curvature (1.1K) | Colored MNIST (10k) |
|---|---|---|
| Vendi Score | 21.9 | 5.8 |
| DCScore | 2.3 | 1.3 |
| MAGAREA | 644.5 | 218.8 |
| PLDiv | 135.2 | 114.3 |
| Sparse PLDiv | 48.0 | 49.0 |
| **Sparse PLDiv (Closed Form)** | **8.2** | **15.7** |

Table 10: Sparse estimation results vs. full matrix results for the Colored MNIST experiment

| Subset | Sparse PLDiv ($\epsilon = 0.3$) | Sparse PLDiv ($\epsilon = 0.8$) | Full Matrix |
|---|---|---|---|
| 1 | 0.45 | 0.45 | 0.45 |
| 2 | 0.77 | 0.77 | 0.77 |
| 3 | 1.17 | 1.18 | 1.17 |
| 4 | 1.48 | 1.48 | 1.48 |
| 5 | 1.86 | 1.86 | 1.86 |
| 6 | 2.09 | 2.09 | 2.09 |
| 7 | 2.46 | 2.47 | 2.46 |
| 8 | 2.88 | 2.89 | 2.88 |
| 9 | 3.32 | 3.33 | 3.33 |
| 10 | 3.69 | 3.69 | 3.69 |

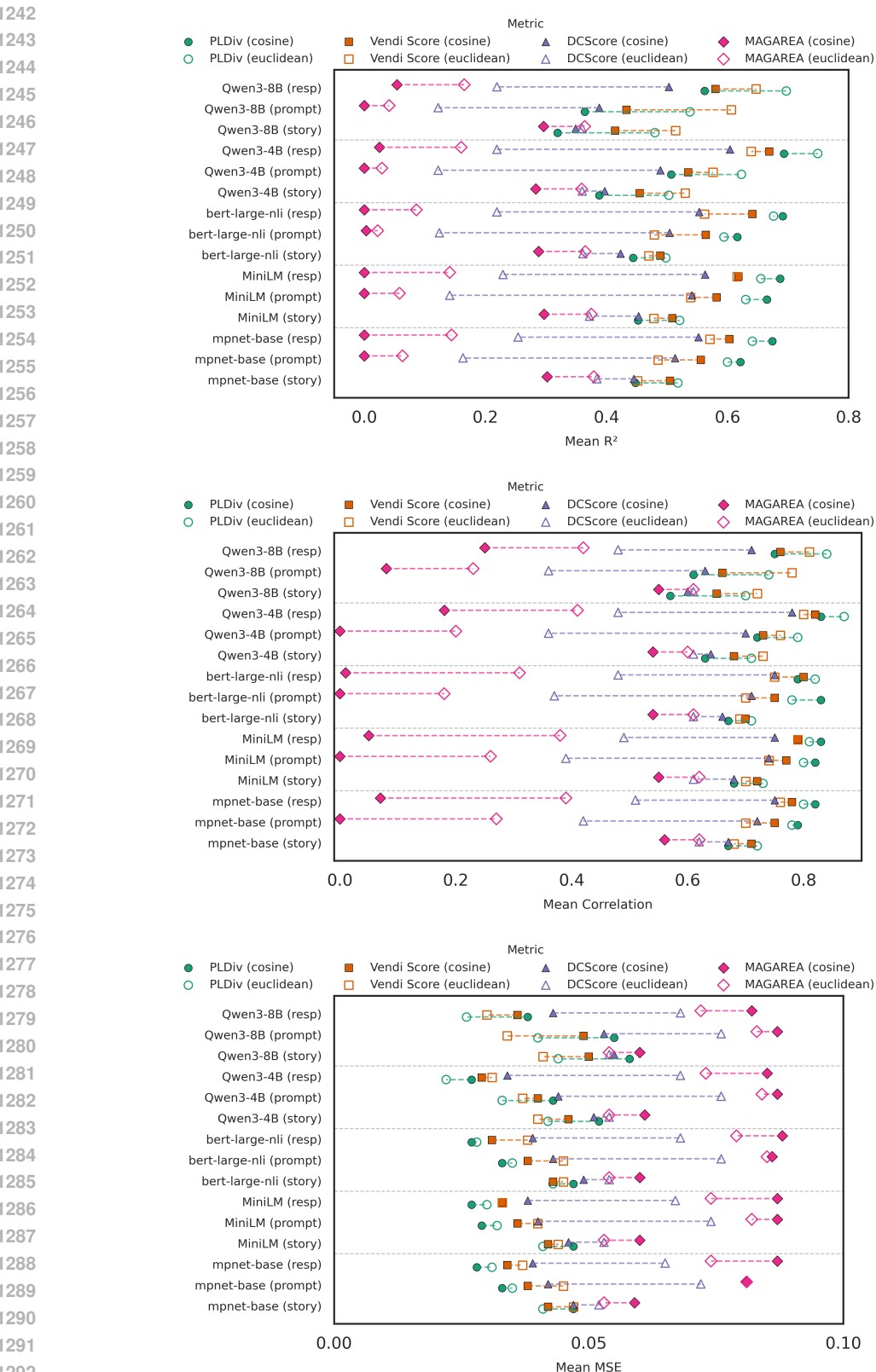

Figure 10: Diversity metric performance is evaluated across different distance/similarity matrices. For Vendi Score and DCScore, the Euclidean setting corresponds to the RBF kernel. PLDiv consistently and reliably outperforms other metrics across various embedding models and distance matrices.

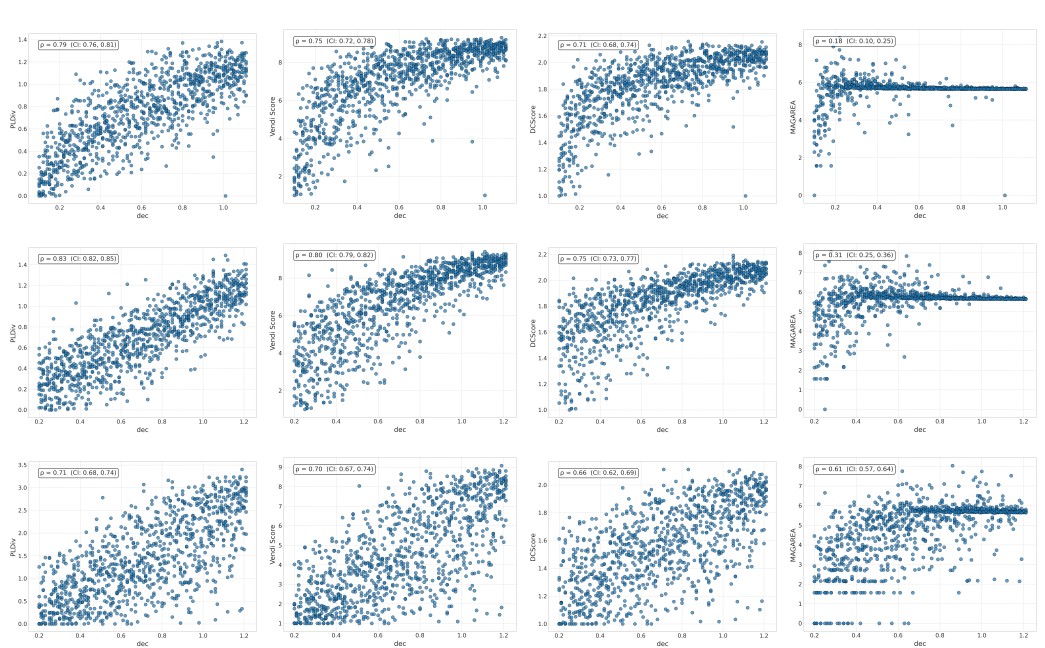

Figure 11: Correlation results for embeddings model: "bert-large-nli-stsb-mean-tokens" across three tasks: Row 1 shows prompt, Row 2 shows response, and Row 3 shows story. Columns 1–4 represent the results for PLDiv, VS, DCS, and MagArea, respectively.

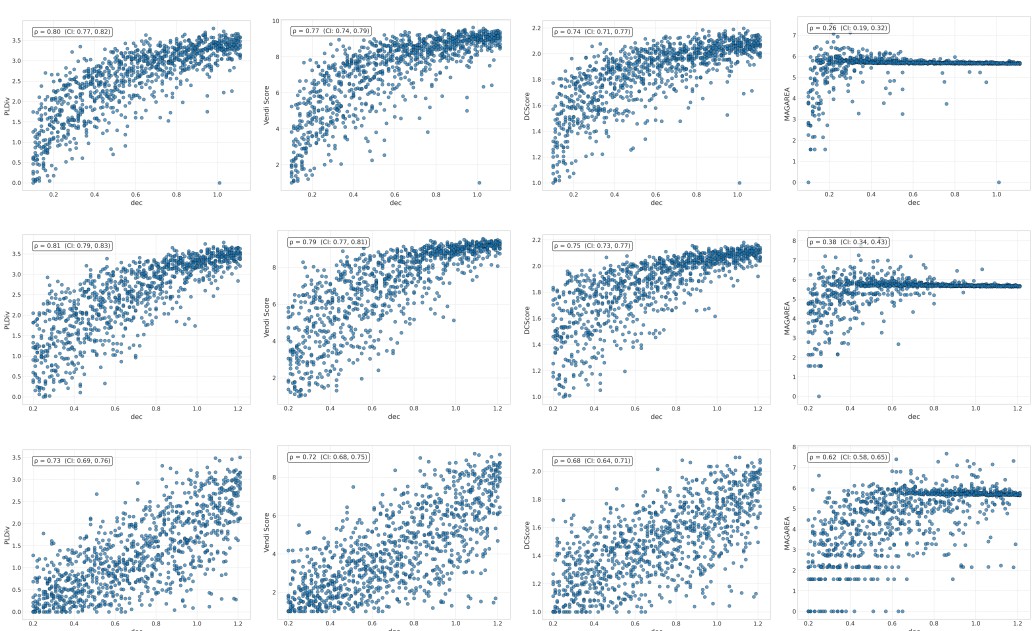

Figure 12: Correlation results for embeddings model: "all-MiniLM-L12-v2" across three tasks: Row 1 shows prompt, Row 2 shows response, and Row 3 shows story. Columns 1–4 represent the results for PLDiv, VS, DCS, and MagArea, respectively.

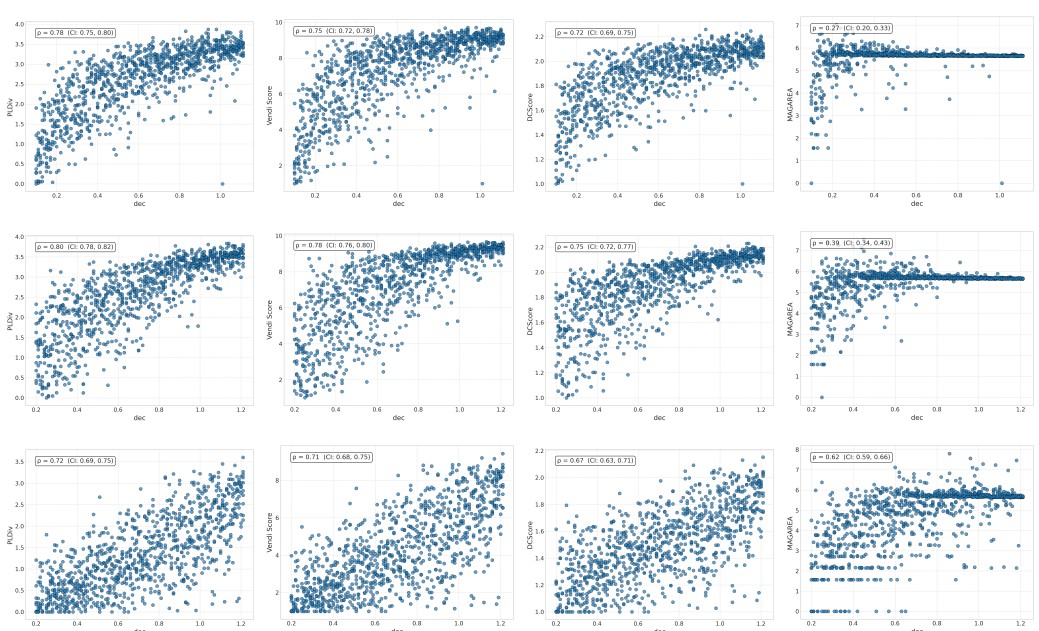

Figure 13: Correlation results for embeddings model: "all-mpnet-base-v2" across three tasks: Row 1 shows prompt, Row 2 shows response, and Row 3 shows story. Columns 1–4 represent the results for PLDiv, VS, DCS, and MagArea, respectively.

