# OpenReview forum: "Geometry-Aware Metric for Dataset Diversity via Persistence Landscapes"
_ICLR.cc/2026/Conference — Submitted to ICLR 2026_

### Official Review · Reviewer_62tN · 2025-10-28

**Soundness:** 2
**Presentation:** 2
**Contribution:** 2
**Rating:** 4
**Confidence:** 3

**Summary:**

This submission focuses on the diversity measure of datasets, which is broadly defined as the presence of meaningful variation across elements. Existing methods primarily consider distributional variation or entropy. This submission futher considers the geometric structure of datasets. To this end, a framework (PLDiv) ) based on topological data analysis (TDA) and persistence landscapes (PLs) is used to characterize the geometric structure. PLDiv is able to link data diversity to its underlying geometry as shown in the experiments.

**Strengths:**

+ It is useful to consider the geometric structure for the diversity measure. This submission uses the topological data analysis (TDA) and persistence landscapes (PLs) to reflect geometry

+ PLDiv has several mathematical properties: it satisfies key diversity axioms (effective size, monotonicity, twin property, symmetry)

**Weaknesses:**

- [**Validation of Geometry-awareness**] The experiments include image embeddings of Colored MNIST and text embeddings of prompt tasks based on MiniLM embeddings. However, there’s no large-scale empirical validation on diverse real-world datasets (e.g., ImageNet variants, multimodal data, or foundation model embeddings).

- [**Comparison with prior metrics**] First, the experiments include Vendi Score, DCScore, and MAGAREA; it does not clearly demonstrate the new information offered by PLDiv. Second, according to Fig. 5, it is hard to see the advantage of PLDiv over other methods. Also, the subset selection in Fig. 3 needs clarification on the advantage of PLDiv

- [**Discussion on sensitivity**] PLDiv depends on distance metrics and filtration parameters, but the paper lacks an analysis of its sensitivity to noise, scaling, or feature normalization

- [**High computational cost**] According to Table 2, PLDiv remains significantly slower than simpler alternatives like Vendi Score or DCScore. Then, there is a need to discuss the scalability of PLDiv, especially for large-scale and high-dimensional datasets.

**Questions:**

- The paper claims PLDiv can guide dataset design and evaluation, but provides no quantitative link between PLDiv scores and downstream model performance or robustness.

- The paper does not include a discussion of its own limitations. Please discuss the scenarios where PLDiv may fail or provide unreliable results. For example, in high-dimensional sparse embeddings or when data points are highly correlated?

---

> ### Author Response · Authors · 2025-11-24
>
> Thank you for reviewing our paper. Below are our answers to your concerns.
>
> > Weakness 1. [Validation of Geometry-awareness] The experiments include image embeddings of Colored MNIST and text embeddings of prompt tasks based on MiniLM embeddings. However, there’s no large-scale empirical validation on diverse real-world datasets (e.g., ImageNet variants, multimodal data, or foundation model embeddings).
>
> As suggested, we have included two LLM-based embedding models (Qwen-Embedding-4B and Qwen-Embedding-8B) in the text embedding experiments. The results indicate that PLDiv performs even better with these advanced high-dimensional embedding models (see the updated Figure 4). Additionally, we sampled ImageNet-1K as a benchmark to compare computation time, as presented in Section 5.6 (Tables 2 and 3). We varied sample sizes of 5k, 10k, 20k, 30k, and 40k, and used ResNet-50 embeddings with cosine similarity/distance, each repeated five times to estimate uncertainty. Our sparse method performs comparably to other methods when the sample size is below 10k. However, as the sample size increases, even the standard PLDiv surpasses the Vendi Score in performance
>
> > Weakness 2. [Comparison with prior metrics] First, the experiments include Vendi Score, DCScore, and MAGAREA; it does not clearly demonstrate the new information offered by PLDiv. Second, according to Fig. 5, it is hard to see the advantage of PLDiv over other methods. Also, the subset selection in Fig. 3 needs clarification on the advantage of PLDiv.
>
> We have updated Section 5.3 (Figure 4) to more clearly demonstrate PLDiv’s superior predictive ability when using both temperature and human-annotated scores as ground truth. In Figure 5, PLDiv clearly outperforms Vendi Score and DCScore, while its advantage over MagArea is less pronounced (correlation: PLDiv = 0.99 vs. MagArea = 0.98). The detailed correlation results are reported in Table 6, Appendix D4.4. Furthermore, we have added a more comprehensive experimental comparison in Section 5 to highlight the unique advantages of PLDiv that other metrics fail to capture (see Section 5.5).
>
> > Weakness  3. [Discussion on sensitivity] PLDiv depends on distance metrics and filtration parameters, but the paper lacks an analysis of its sensitivity to noise, scaling, or feature normalization.
>
> We have included a small experiment in Appendix D.1.1 to demonstrate PLDiv’s performance on imbalanced and small-perturbation data. Section 5.5 further presents several scenarios where PLDiv effectively captures subtle differences. Theoretically, PLDiv is robust to noise, as demonstrated by the sparse distance matrix method, which shows that such perturbations do not affect the final results (see Table 3). We do not recommend normalization for the cosine distance matrix, but we suggest applying it when using the Euclidean distance matrix, as we stated in the experiment description.  Moreover, we explore how different distance matrices/kernels impact the diversity metric performance. Figure 10 in Appendix D.4 to illustrate that PLDiv remains robust across different distance matrices.
>
> > Weakness  4. [High computational cost] According to Table 2, PLDiv remains significantly slower than simpler alternatives like Vendi Score or DCScore. Then, there is a need to discuss the scalability of PLDiv, especially for large-scale and high-dimensional datasets.
>
> We have revised Section 5.6 (Computational Complexity) to provide a more comprehensive comparison and analysis. Sample sizes of 5k, 10k, 20k, 30k, and 40k were drawn from ImageNet-1K, and experiments were conducted using ResNet-50 embeddings with cosine similarity/distance, each repeated five times to estimate uncertainty. Our sparse method performs comparably to other methods when the sample size is below 10k. However, as the sample size increases, even the standard PLDiv surpasses the Vendi Score in performance. Please refer to Table 2 in the revised manuscript for detailed results.
>
> > Question 1. The paper claims PLDiv can guide dataset design and evaluation, but provides no quantitative link between PLDiv scores and downstream model performance or robustness.
>
> We appreciate this observation. Our primary goal in this paper is to introduce and validate PLDiv as a theoretically grounded and geometry-aware measure of diversity, rather than to establish its impact on all possible downstream tasks it may benefit. The sentence in the conclusion was intended to suggest potential applications of PLDiv, which we plan to explore in future research. We have also revised this sentence to avoid any confusion.
>
> > Question 2. The paper does not include a discussion of its own limitations. Please discuss the scenarios where PLDiv may fail or provide unreliable results. For example, in high-dimensional sparse embeddings or when data points are highly correlated?
>
> Thanks for raising this concern. We've added a limitations section in Appendix E.

---

### Official Review · Reviewer_Qxac · 2025-10-30

**Soundness:** 2
**Presentation:** 2
**Contribution:** 2
**Rating:** 4
**Confidence:** 5

**Summary:**

The paper proposes PLDiv, a new diversity measure based on persistent homology (PH), which aims to quantify diversity in data from a geometric and topological perspective. By using 0-dimensional persistence landscapes, PLDiv captures clustering behaviour and satisfies four axioms of diversity. PLDiv is computed for synthetic, image, and text datasets. Results show the potential of PLDiv for capturing geometric information by (i) distinguishing simulated point patterns, (ii) predicting curvature, (iii) measuring the diversity of generated text as determined by the softmax temperature, and (iv) correlating with the number of unique classes sampled from an image dataset when computed from image embeddings. Further, empirical runtimes are compared to alternative diversity measures.

**Strengths:**

- The text is clearly written and the main question it addresses on how to measure diversity is relevant in application.

- The application of persistent homology (PH) to diversity evaluation for ML tasks is novel and theoretically interesting.

- The proposed summary, PLDiv, is geometry-aware and measures a notion of diversity related to 0-dimensional PH features e.g. clustering behaviour.

- The proposed diversity measure is theoretically motivated and fulfills four basic axioms of diversity.

- Evaluation across multiple data modalities (synthetic, image, text) demonstrates the flexibility of the proposed diversity measure.

- It is beneficial that uncertainty is evaluated for some of the results (Table 1 and Figure 4) and that a sparsification approach is considered to speed-up PH computations.

**Weaknesses:**

The paper’s empirical and conceptual claims are not fully substantiated, which limits its impact.

**Overstated Claims about Geometry-Awareness:** The paper claims that existing diversity measures “neglect geometric structure”, or do not "genuinely consider data from a geometric perspective". However, measures like VS, MagArea, and MagDiff can be computed from the same pairwise distance matrices used to calculate PLDiv and are inherently geometry-dependent. Exactly how geometry is summarised by either of these diversity measures is a more nuanced debate that requires clearer formalisation or further empirical investigation.

**Limited Advantages over Baselines:** The practical benefits of PLDiv over existing diversity measures, such as the Vendi Score (VS), MagArea, or MagDiff are not convincingly demonstrated (as detailed below). It would be of interest to show stronger examples of diversity evaluation tasks that uniquely require PLDiv and cannot be addressed by established measures.

**Point Pattern Analysis:**  Figure 3 shows that PLDiv can distinguish between simulated point patterns of varying diversity. However, it is not evaluated  if PLDiv has an unique advantage at this task or if other diversity measures could also tell the difference between these examples.

**Curvature prediction (Table 1):** VS and MagArea can predict curvature with MSEs of 0.05 compared to PLDiv with an MSE of 0.04 (see Limbeck et al. and Turks et al. for comparison). Given the experimental setup, and the small absolute difference in MSE, this does not seem like conclusive evidence that PLDiv is uniquely geometry-aware. For further clarifications, it would also be of interest to simply report the values of each diversity score plotted against the curvature values.

**Text Evaluation (Figures 4, 6, 7,8):** Considering uncertainties, all diversity measures reach similar Spearman correlations on the response task, and PLDiv shows the lowest Spearman correlation on the story task. The authors state that PLDiv exhibits linear correlation, but if the stronger linear relationship is its main advantage over other diversity measures, Pearson correlation should be evaluated instead. Questions of interest beyond the correlation with the softmax temperature, such as the alignment with human evaluation scores are not evaluated. The reported text evaluation tasks already seem to be sufficiently addressed in more detail by baseline diversity measures (see Zhu et al. or Limbeck et al. for comparison).

**Image Evaluation:** MagArea seems to show very similar trends and perform on-par with PLDiv in Figure 5. To clarify whether PLDiv performs best at this experiment, it would be beneficial to report uncertainties e.g. by repeating the experiment across varying different seeds or subsamples. Further, it would be relevant to specify which parameter and distance choices have been used to compute alternative diversity measures and how these impact results to e.g. discuss why the DCScore here shows larger fluctuations than reported by Zhu et al..

**Limited Scalability:** PLDiv shows e.g. MagArea with the parameter settings reported here. However, the runtimes of PLDiv are not necessarily advantageous over diversity measures that are computed at one fixed scale of (dis)similarity e.g. the VS score, which is still notably faster than PLDiv for the examples in Table 2..

**Reproducibility:** Questions remain on the reproducibility of the experimental results. A reproducibility section and relevant supplementary materials detailing the specific experimental setups and parameter choices would improve transparency.

**Theoretical Novelty:** Existing entropy-based diversity measures are also theoretically justified and the authors state no unique theoretic property of PLDiv that is not also already fulfilled by an alternative diversity measure.

**Overall Assessment:** This paper introduces a valid and mathematically sound approach to measure diversity via persistent homology. However, the claimed advantages over existing diversity measures are not sufficiently substantiated, both conceptually and empirically. The work would be strengthened by more careful positioning relative to prior methods, clearer definitions of what “geometry-aware” means in this context, and broader, reproducible experiments that highlight for exactly which datasets PLDiv provides unique insights.

**Questions:**

**Main Questions:**

- Line 139 claims that magnitude-based methods "abstract away the geometric or topological structures that can differentiate datasets with the same dispersion."
Can the authors provide examples where PLDiv distinguishes datasets that alternative diversity measures cannot?

- Does PLDiv fulfill any theoretic properties relating to diversity (beyond what is shown in 4.2.) that are not fulfilled by other diversity measures ?

- Why is PLDiv proposed as a new summary of PH rather than applying existing summary statistics? Would the trends shown by PLDiv be different to e.g. total persistence or other one-number summaries of PH? Further comparison could be relevant to clearly motivate the introduction of PLDiv.

- Is PLDiv (sufficiently) scalable to large datasets compared to e.g. the Vendi Score? Further empirical evaluation on increasing sample sizes would be of interest.

- Could PLDiv also be generalised via alternative filtrations (e.g., using cosine distances) to capture structure in other modalities such as language embeddings?  Would being similarity-dependent then not be a strength rather than a limitation?

- Which implementations of the alternative diversity measures, dissimilarity choices, and other parameter choices have been used to report the results in Table 2, Figure 4 or Figure 5? Does the choice of dissimilarity impact results and does it ensure a fair comparison?

---

**Further Questions on Reproducibility:**

- How is the k-DPP sampling implemented and can you cite a reference?

- How exactly is the experiment in Section 5.4. simulated? Does the number of observations stay the same or does it decrease as the number of labels decreases?

- Do you have an intuition on why PLDiv shows the lowest standard deviation in the MSE in Table 1?

- Missing implementation details are given as a reason for not reporting MagArea in Figure 4. Why could MagArea be computed for other experiments but not for this one? Could it not be calculated from e.g. the same distances used to calculate PLDiv?

- Which dataset is Table 5 computed on?

- Line 062 cites Bubenik et al. directly after the statement that "curvature is inherently linked to diversity", but their paper never once mentions diversity. Isn’t the statement taken from another reference?

- Conclusions state that “these results establish PLDiv as a versatile tool for dataset construction, augmentation, model evaluation, and robustness analysis”. But as far as I am aware, dataset construction, augmentation, and robustness analysis are not explicitly analysed in the experiments?

---

> ### Author Response · Authors · 2025-11-24
>
> Thanks for reviewing our work. Our responses are provided below.
>
> > Weakness 1. Overstated Claims about Geometry-Awareness: The paper claims that existing diversity measures “neglect geometric structure”, or do not "genuinely consider data from a geometric perspective". However, measures like VS, MagArea, and MagDiff can be computed from the same pairwise distance matrices used to calculate PLDiv and are inherently geometry-dependent. Exactly how geometry is summarised by either of these diversity measures is a more nuanced debate that requires clearer formalisation or further empirical investigation.
>
> We thank the reviewer for this insightful observation and agree that existing metrics rely on pairwise distance matrices and thus incorporate a metric notion of geometry. Our claim is not that these approaches are entirely geometry-agnostic, but that they summarize geometry only indirectly. In contrast, PLDiv derives from persistent homology, which tracks how topological features evolve across scales. This allows PLDiv to explicitly capture geometric structure via topological persistence, yielding a richer and more interpretable characterization of dataset geometry. In our updated manuscript, Section 5.5 concretely illustrates what we mean by this geometry-aware property.
>
> >Weakness 2.  Limited Advantages over Baselines: The practical benefits of PLDiv over existing diversity measures, such as the Vendi Score (VS), MagArea, or MagDiff are not convincingly demonstrated (as detailed below). It would be of interest to show stronger examples of diversity evaluation tasks that uniquely require PLDiv and cannot be addressed by established measures.
>
> Our experiments in Sections 5.2, 5.3, and 5.4 consistently demonstrate that our proposed metric outperforms existing diversity measures across different modalities. In addition, we have included an additional experiment (Section 5.5) to illustrate more scenarios where PLDiv excels, highlighting its geometry-aware capability that other metrics lack.
>
> >Weakness 3. Point Pattern Analysis: Figure 3 shows that PLDiv can distinguish between simulated point patterns of varying diversity. However, it is not evaluated if PLDiv has an unique advantage at this task or if other diversity measures could also tell the difference between these examples.
>
> Thanks for this comment.  We added these values in Appendix D.2.  Please see Table 5.
>
> > Weakness 4. Curvature prediction (Table 1): VS and MagArea can predict curvature with MSEs of 0.05 compared to PLDiv with an MSE of 0.04 (see Limbeck et al. and Turks et al. for comparison). Given the experimental setup, and the small absolute difference in MSE, this does not seem like conclusive evidence that PLDiv is uniquely geometry-aware. For further clarifications, it would also be of interest to simply report the values of each diversity score plotted against the curvature values.
>
> Thanks for pointing this out. As suggested, we provided visualizations of the diversity measures against the curvature labels in Appendix D.3. It shows that PLDiv achieves the best separation between positive and negative curvatures, providing clear evidence of why PLDiv performs best in Table 1. Please refer to Figure 8.
>
> > Weakness 5. Text Evaluation (Figures 4, 6, 7,8): Considering uncertainties, all diversity measures reach similar Spearman correlations on the response task, and PLDiv shows the lowest Spearman correlation on the story task. The authors state that PLDiv exhibits linear correlation, ...  Questions of interest beyond the correlation with the softmax temperature, such as the alignment with human evaluation scores are not evaluated. The reported text evaluation tasks already seem to be sufficiently addressed in more detail by baseline diversity measures (see Zhu et al. or Limbeck et al. for comparison).
>
> We respectfully disagree that the text evaluation tasks in Section 5.3 have been sufficiently addressed in previous literature. According to Limbeck et al. (2024), the $R_2$ values remain relatively low (around 0.6) for some tasks, indicating substantial room for improving current diversity measures. We have revised Section 5.3 to include experiments with human-annotated data, two recent LLM-based embedding models (Qwen-4B and Qwen-8B), and additional evaluation metrics such as R², MSE, and Pearson correlation. The updated results show that PLDiv achieves the best performance across tasks and embedding models when using softmax temperature $dec$ as the ground truth, and ranks among the top results when using human annotations (_ABS-HDS_) as labels. Moreover, when higher-dimensional embedding models (Qwen-4B/8B) are applied, the superiority of our metric becomes even more pronounced. In contrast, DCScore (Zhu et al., 2025) shows inferior performance in several experiments, while MagArea (Limbeck et al., 2024) requires substantially more computation time.

---

> > ### Author Response · Authors · 2025-11-24
> >
> > > Weakness: 6. Image Evaluation: MagArea seems to show very similar trends and perform on par with PLDiv in Figure 5. To clarify whether PLDiv performs best at this experiment, it would be beneficial to report uncertainties e.g. by repeating the experiment across varying different seeds or subsamples. Further, it would be relevant to specify which parameter and distance choices have been used to compute alternative diversity measures and how these impact results to e.g. discuss why the DCScore here shows larger fluctuations than reported by Zhu et al.
> >
> > As suggested, we conducted each experiment five times to assess the stability of the results. The outcomes showed minimal variation, with each diversity score remaining consistent across runs. We report the detailed values in Table 6 in Appendix D.5, along with the corresponding correlation scores, demonstrating that our method achieves the best performance.
> >
> > All experiments were conducted using cosine distance/similarity. To investigate the fluctuations observed in DCScore, we evaluated the metric both with and without normalization (the default configuration in their implementation is without normalization). Applying normalization further degraded performance, following a trend similar to that of the Vendi Score. A similar observation was also found in the text evaluation experiment.
> >
> > Based on our analysis, DCScore consistently underperforms compared to other metrics, particularly relative to Vendi Score, not just in image embeddings. Although DCScore is designed as an optimization-based extension of Vendi Score with improved computational efficiency, its accuracy and stability remain inferior.
> >
> > > Weakness 7. Limited Scalability: PLDiv shows e.g. MagArea with the parameter settings reported here. However, the runtimes of PLDiv are not necessarily advantageous over diversity measures that are computed at one fixed scale of (dis)similarity e.g. the VS score, which is still notably faster than PLDiv for the examples in Table 2.
> >
> > We have revised Section 5.6 (Computational Complexity) to provide a more comprehensive comparison and analysis. Sample sizes of 5k, 10k, 20k, 30k, and 40k were drawn from ImageNet-1K, and experiments were conducted using ResNet-50 embeddings with cosine similarity/distance, each repeated five times to estimate uncertainty. Our sparse method performs comparably to other methods when the sample size is below 10k. However, as the sample size increases, even the standard PLDiv surpasses the Vendi Score in performance. Please refer to Table 2 in the revised manuscript for detailed results.
> >
> > > Weakness 8. Reproducibility: Questions remain on the reproducibility of the experimental results. A reproducibility section and relevant supplementary materials detailing the specific experimental setups and parameter choices would improve transparency.
> >
> > Thank you for raising this concern. We have added a detailed description of the experimental implementation in Appendix D and have also provided our code in the supplementary material for reference.
> >
> > > Weakness 9. Theoretical Novelty: Existing entropy-based diversity measures are also theoretically justified and the authors state no unique theoretic property of PLDiv that is not also already fulfilled by an alternative diversity measure.
> >
> > We respectfully disagree that existing metrics are sufficient. Our experiments consistently show that PLDiv outperforms alternative metrics across multiple settings and data modalities. Even at the axiomatic level, some existing measures only partially satisfy fundamental diversity properties. For example, while the Vendi Score claims to satisfy monotonicity, both Limbeck et al. 2024 and our experiments show that these measures are insensitive to label richness, thereby violating the monotonicity property in practice.

---

> > > ### Author Response · Authors · 2025-11-24
> > >
> > > > Weakness: 10. Overall Assessment: This paper introduces a valid and mathematically sound approach to measure diversity via persistent homology. However, the claimed advantages over existing diversity measures are not sufficiently substantiated, both conceptually and empirically. The work would be strengthened by more careful positioning relative to prior methods, clearer definitions of what “geometry-aware” means in this context, and broader, reproducible experiments that highlight for exactly which datasets PLDiv provides unique insights.
> > >
> > > Thanks for acknowledging the novelty and soundness of our proposed approach. Our main contribution is that PLDiv bridges geometric structure and statistical diversity. It satisfies four core diversity principles both theoretically and empirically, and consistently outperforms alternative metrics across a range of applications and data modalities. In Section 5.5, we introduce an additional experiment demonstrating that PLDiv can effectively capture geometric structures where existing diversity metrics fail. Although computational efficiency is not our primary focus, experiments on ImageNet-1K show that PLDiv scales well to large-scale datasets and is more efficient than Vendi Score and MagArea.
> > >
> > >
> > > > Main Questions 1. Line 139 claims that magnitude-based methods "abstract away the geometric or topological structures that can differentiate datasets with the same dispersion." Can the authors provide examples where PLDiv distinguishes datasets that alternative diversity measures cannot?
> > >
> > > We thank the reviewer's suggestion. We added a new experiment in Section 5.5 to address this concern. We simulated eight pairs of two-dimensional point clouds (A, B), each containing about 200 points generated from parameterized geometric functions described in the Appendix Table 7. Our metric, PLDiv, consistently outperforms other metrics across all test cases, capturing geometrically meaningful and visually intuitive differences between each pair of point clouds more effectively. Please refer to Section  5.5 and Figure 6.
> > >
> > > > Main Questions 2. Does PLDiv fulfill any theoretic properties relating to diversity (beyond what is shown in 4.2.) that are not fulfilled by other diversity measures ?
> > >
> > > Thanks for your comments. However, we don't believe additional properties bring additional benefits. Leinster (2012) organizes diversity properties into three major categories: "partitioning properties", "elementary properties", and "effects of species similarity on diversity", each comprising several subcomponents. Among these, our metric satisfies four key properties distributed across the three categories. Other subcomponents either implicitly encompass these four properties or are not directly applicable in the context of machine learning datasets, where the interpretation of “species” and “similarity” differs from ecological settings. Based on our experiments, we also demonstrate that certain other metrics, such as the Vendi Score, fail to fully satisfy some of the required axioms where our metric meets those properties, as demonstrated in Figures 1, 5, and 6.

---

> > > > ### Author Response · Authors · 2025-11-24
> > > >
> > > > > Main Questions 3. Why is PLDiv proposed as a new summary of PH rather than applying existing summary statistics? Would the trends shown by PLDiv be different to e.g. total persistence or other one-number summaries of PH? Further comparison could be relevant to clearly motivate the introduction of PLDiv.
> > > >
> > > > Thanks for your question. If we review PLDiv from the closed form, there seem to be other summaries of PH choices. But we would like to note that the closed form is not manually designed; instead, it comes from the definition of PLDiv (persistence landscapes). We've stated our motivation for PLDiv in Section 4.1.
> > > >
> > > > Regarding the reason why we utilize persistence landscapes rather than persistent homology directly, the key advantage lies in the fact that landscapes provide a functional and linear representation of the topological information, which allows for straightforward computation, averaging, and integration within standard statistical and machine learning frameworks. Additionally, persistence landscapes possess a desirable stability property (Bubenik, P.,  2015), meaning that small perturbations or noise in the input data that slightly shift birth and death times in the persistence diagram lead to only small, controlled changes in the resulting landscape under the $L^p$ norm. Consequently, this stability naturally carries over to our proposed diversity measure, ensuring that PLDiv is also robust to noise and comparable across datasets, while maintaining a clear geometric interpretation as the total area under the persistence landscape.
> > > >
> > > > Not all Summaries of PH would satisfy all the diversity axioms. For instance, the mean or the entropy of lifetimes may violate monotonicity or other desired properties. The total lifetime satisfies the core axioms, but the second-order moment (quadratic function of lifetimes) offers distinct advantages. Specifically, the quadratic form emphasizes long-lived topological features while suppressing short-lived, noisy ones, making PLDiv more robust to small perturbations. Moreover, PLDiv is differentiable, which facilitates its integration into machine learning frameworks for optimization or gradient-based learning. Finally, because it represents a second-moment measure, PLDiv is closely related to the notion of variance in statistics—commonly interpreted as a measure of diversity or dispersion. The derived closed form makes this connection explicit, reinforcing the theoretical soundness and distributional interpretability of PLDiv as a diversity measure.
> > > >
> > > >
> > > > > Main Questions 4. Is PLDiv (sufficiently) scalable to large datasets compared to e.g. the Vendi Score? Further empirical evaluation on increasing sample sizes would be of interest.
> > > >
> > > > We have revised Section 5.6 (Computational Complexity) to provide a more comprehensive comparison and analysis. Sample sizes of 5k, 10k, 20k, 30k, and 40k were drawn from ImageNet-1K, and experiments were conducted using ResNet-50 embeddings with cosine similarity/distance, each repeated five times to estimate uncertainty. Our sparse method performs comparably when the sample size is below 10k. However, as the sample size increases, even the standard PLDiv surpasses the Vendi Score in performance. Please refer to Table 2 in the revised manuscript for detailed results.
> > > >
> > > > > Main Questions 5. Could PLDiv also be generalised via alternative filtrations (e.g., using cosine distances) to capture structure in other modalities such as language embeddings? Would being similarity-dependent then not be a strength rather than a limitation?
> > > >
> > > > Our method can operate on any given distance matrix. When a data cloud is provided instead of a distance matrix, persistent homology internally computes Euclidean distances by default. Currently, PLDiv employs the Vietoris–Rips filtration, but in future work, we plan to explore alternative filtrations (e.g., Čech or Alpha filtrations). Similarity-dependent metrics are not a limitation. Each similarity function implicitly corresponds to a distance metric. For example, the RBF kernel is related to Euclidean distance, and the Laplacian kernel reflects Manhattan distance. Therefore, being similarity-dependent simply defines the geometric space in which relationships are measured, rather than restricting the method.

---

> > > > > ### Author Response · Authors · 2025-11-24
> > > > >
> > > > > > Main Questions 6. Which implementations of the alternative diversity measures, dissimilarity choices, and other parameter choices have been used to report the results in Table 2, Figure 4 or Figure 5? Does the choice of dissimilarity impact results and does it ensure a fair comparison?
> > > > >
> > > > > We experiment by using the same distance matrix or kernel and parameter setting. Section 5.4 employs cosine distance/similarity, while Section 5.5 uses Euclidean distance. In Section 5.3, we use Euclidean distance for PLDiv and MagArea, and cosine similarity (as implemented in their original code by default) for Vendi Score and DCScore. Notably, when switching Vendi Score and DCScore to Euclidean distance, their performance degrades even further. We provide our code and experimental details, along with a visualization (Figure 10) in Appendix D.
> > > > >
> > > > > > Further Questions 1. How is the k-DPP sampling implemented and can you cite a reference?
> > > > >
> > > > > Thanks for pointing this out. We provided a detailed description of it in Appendix D.2 and added the citation as well.
> > > > >
> > > > > > Further Questions 2. How exactly is the experiment in Section 5.4. simulated? Does the number of observations stay the same or does it decrease as the number of labels decreases?
> > > > >
> > > > > Section 5.4 presents experiments on ColoredMNIST, which contains 10 classes. We sampled half of the data from each class. Starting from class 1, we incrementally added samples from one additional class at a time, up to class 10, thereby forming 10 subsets. Therefore, the sample size decreases as the number of labels decreases.
> > > > >
> > > > > > Further Questions 3. Do you have an intuition on why PLDiv shows the lowest standard deviation in the MSE in Table 1?
> > > > >
> > > > > The lowest standard deviation of PLDiv indicates that our metric can better capture geometric information in the curvature data more effectively. In our experiment, we kept the parameter settings the same for all metrics and applied 5-fold cross-validation. The 2D curvature was used directly (which internally computes Euclidean distances). We detailed the experiment and plotted each metric against the curvature labels in Appendix 5.3.  Figure 8 provides evidence that PLDiv achieves the best separation between positive and negative curvatures and may answer the question with the lowest standard deviation.
> > > > >
> > > > >
> > > > > > Further Questions 4. Missing implementation details are given as a reason for not reporting MagArea in Figure 4. Why could MagArea be computed for other experiments but not for this one? Could it not be calculated from e.g. the same distances used to calculate PLDiv?
> > > > >
> > > > > MagArea outputs didn't converge for $resp$ and $prompt$ tasks. We use the same distance matrix as PLDiv and the same parameter that worked well for $story$ tasks, but failed to produce meaningful outcomes for the other two. We have updated Figure 4 to include MagArea for comparison and added a correlation plot in Figures 11, 12, and 13 to visualize its outputs. Furthermore, we added a section in Appendix D.4 to detail experimental implementation and uploaded code in the supplementary material.
> > > > >
> > > > >
> > > > > > Further Questions 5. Which dataset is Table 5 computed on?
> > > > >
> > > > > In our original manuscript, there is no Table 5; we assume you are referring to Table 3, which reports the experiment on ColoredMNIST (described in Section 5.4). To provide a more comprehensive large-scale dataset analysis (ImageNet-1K), we have now moved this table to the Appendix as Table 10.
> > > > >
> > > > >
> > > > > > Further Questions 6. Line 062 cites Bubenik et al. directly after the statement that "curvature is inherently linked to diversity", but their paper never once mentions diversity. Isn’t the statement taken from another reference?
> > > > >
> > > > > Sorry for the confusion. It comes from "Limbeck, K., Andreeva, R., Sarkar, R., & Rieck, B. (2024). Metric space magnitude for evaluating the diversity of latent representations. Advances in Neural Information Processing Systems", where they cite Bubenik et al., 2020. We have updated the citation.
> > > > >
> > > > > > Further Questions 7. Conclusions state that “these results establish PLDiv as a versatile tool for dataset construction, augmentation, model evaluation, and robustness analysis”. But as far as I am aware, dataset construction, augmentation, and robustness analysis are not explicitly analysed in the experiments?
> > > > >
> > > > > We appreciate this observation. Our primary goal in this paper is to introduce and validate PLDiv as a theoretically grounded and geometry-aware measure of diversity, rather than to establish its impact on all possible downstream tasks it may benefit. The sentence in the conclusion was intended to suggest potential applications of PLDiv, which we plan to explore in future research. We have also revised this sentence to avoid any confusion.

---

### Official Review · Reviewer_jSSq · 2025-10-31

**Soundness:** 2
**Presentation:** 3
**Contribution:** 2
**Rating:** 4
**Confidence:** 4

**Summary:**

The paper proposes a new dataset diversity metric, PLDiv, based on topological data analysis (TDA) and persistence landscapes. The authors argue that existing diversity measures—such as entropy, magnitude, or kernel-based metrics—fail to capture geometric structures in the data. By leveraging persistent homology and integrating the resulting “tent” functions, PLDiv aims to provide a geometry-aware quantification of diversity. The paper claims theoretical justification (via diversity axioms) and reports experiments across synthetic data, text embeddings, and image embeddings, suggesting that PLDiv correlates with intuitive diversity notions.

**Strengths:**

1). The topic of dataset diversity measurement is important and increasingly relevant for evaluating generative models and representation learning.

2). The exposition is clear and the paper is well written overall, with some effort spent on theoretical motivation.

**Weaknesses:**

1). Lack of substantial novelty. The proposed PLDiv essentially computes a quadratic function of the lifetimes in persistent homology (𝑃𝐿𝐷𝑖𝑣=1/4∑𝑖(𝑑𝑖−𝑏𝑖)^2). This is a straightforward statistic derived from standard persistence diagrams, not a fundamentally new methodology. The connection to diversity measurement appears superficial, and the theoretical content (axioms, proofs) largely restates basic TDA properties rather than introducing new insights.

2). Weak theoretical contribution. The “axiomatic” analysis simply verifies trivial properties (monotonicity, symmetry, etc.) that hold for any nonnegative function of lifetimes. There is no rigorous justification that PLDiv truly measures diversity in a meaningful or generalizable sense, beyond restating intuition.

3). Empirical evaluation lacks depth and rigor. The experimental settings are small-scale and largely qualitative. Comparisons with existing metrics (Vendi Score, DCScore, MAGAREA) rely on correlation or regression but lack clear baselines, statistical tests, or ablation analyses. There is no real-world application where PLDiv produces actionable improvements or insights.

4). The method requires persistent homology computations over the full distance matrix, which is expensive for large datasets. The “sparse” approximation in Table 2 still consumes considerable time without clear scalability to modern data scales.

**Questions:**

See the weaknesses.

---

> ### Author Response · Authors · 2025-11-24
>
> Thanks for your review. Let us clarify your concerns and, maybe, some misunderstandings.
>
> >1. Lack of substantial novelty. The proposed PLDiv essentially computes a quadratic function of the lifetimes in persistent homology ($PLDiv = 1/4 \sum (d_i - b_i)^2$). This is a straightforward statistic derived from standard persistence diagrams, not a fundamentally new methodology. The connection to diversity measurement appears superficial, and the theoretical content (axioms, proofs) largely restates basic TDA properties rather than introducing new insights.
>
> We respectfully disagree with the claim that the simple and elegant statistic is equivalent to non-novelty; instead, it's an advantage. The closed form of PLDiv is not manually designed; rather, it is derived from the definition of PLDiv (from persistence landscapes), making it a result rather than a starting assumption.
>
> Regarding the reason why we utilize persistence landscapes rather than persistent homology directly, the key advantage lies in the fact that landscapes provide a functional and linear representation of the topological information, which allows for straightforward computation, averaging, and integration within standard statistical and machine learning frameworks. Additionally, persistence landscapes possess a desirable stability property (Bubenik, P.,  2015), meaning that small perturbations or noise in the input data that slightly shift birth and death times in the persistence diagram lead to only small, controlled changes in the resulting landscape under the $L^p$ norm.  Consequently, this stability naturally carries over to our proposed diversity measure, ensuring that PLDiv is also robust to noise and comparable across datasets, while maintaining a clear geometric interpretation as the total area under the persistence landscape.
>
> The second-order moment is closely related to the “variance” of a variable, which is generally interpreted as a measure of diversity from a distributional perspective. Our derived closed form connects to this concept, demonstrating the theoretical soundness of PLDiv. Additionally, the quadratic form emphasizes long-lived topological features while suppressing short-lived, noisy ones, making PLDiv more robust to small perturbations. We've indicated this in remarks 4.4 and 4.5. Also, PLDiv is differentiable, which facilitates its integration into machine learning frameworks for optimization or gradient-based learning.
>
> >2. Weak theoretical contribution. The “axiomatic” analysis simply verifies trivial properties (monotonicity, symmetry, etc.) that hold for any nonnegative function of lifetimes. There is no rigorous justification that PLDiv truly measures diversity in a meaningful or generalizable sense, beyond restating intuition.
>
> The axioms proposed by Leinster \& Cobbold (2012) provide an important theoretical lens for evaluating whether a diversity metric aligns with fundamental and intuitive properties of diversity. In line with the prior work in the diversity literature, in which most measures are validated against four such axioms, we formally prove that PLDiv satisfies four key axioms: effective size, monotonicity, twin property, and symmetry.
>
> The axioms, such as effective size and monotonicity, are not only theoretically proven in Section 4.2 but are also empirically verified through our toy synthetic examples and image embedding experiments in Section 5.4. We further include additional experiments in Section 5.5 to demonstrate that PLDiv consistently serves as a reliable and geometry-aware metric for quantifying dataset diversity.
>
> We would like to note that the reviewer’s claim that “any nonnegative function of lifetimes satisfies these properties” is not correct. For example, the mean lifetime does not satisfy the effective size axiom: adding more well-separated points can leave the average unchanged, violating the intuition that diversity should increase. Similarly, persistent entropy may even decrease when the distribution of lifetimes changes unevenly.
>
> As stated in our response to weakness 1, PLDiv is not an arbitrary nonnegative function: it originates from the persistence landscape. Its closed formulation is not merely a mathematical convenience but a consequence of topological persistence, linking interpretability, robustness, and geometric meaning in a principled way.
>
> "Bubenik, P.,  2015" : Bubenik, P. (2015). Statistical topological data analysis using persistence landscapes. _J. Mach. Learn. Res._, _16_(1), 77-102.

---

> ### Author Response · Authors · 2025-11-24
>
> > 3. Empirical evaluation lacks depth and rigor. The experimental settings are small-scale and largely qualitative. Comparisons with existing metrics (Vendi Score, DCScore, MAGAREA) rely on correlation or regression but lack clear baselines, statistical tests, or ablation analyses. There is no real-world application where PLDiv produces actionable improvements or insights.
>
> In existing metrics (Vendi Score, DCScore, MAGAREA), the analyses primarily rely on correlation and regression methods. Since our paper does not aim to improve AI model architecture, baseline or ablation experiments are not applicable in this context. Instead, we compare our proposed metrics with the existing ones. Furthermore, we have revised Section 5.3 to include $R^2$ and MSE comparisons, providing more quantitative evaluation results. We also add a large-scale dataset experiment on ImageNet-1K in Section 5.6.
>
> >4. The method requires persistent homology computations over the full distance matrix, which is expensive for large datasets. The “sparse” approximation in Table 2 still consumes considerable time without clear scalability to modern data scales.
>
> Thanks for pointing this out. We have revised Section 5.6 (Computational Complexity) to provide a more comprehensive comparison and analysis. Sample sizes of 5k, 10k, 20k, 30k, and 40k were drawn from ImageNet-1K, and experiments were conducted using ResNet-50 embeddings with cosine similarity/distance, each repeated five times to estimate uncertainty. Our sparse method performs comparably to other methods when the sample size is below 10k. However, as the sample size increases, even the standard PLDiv surpasses the Vendi Score in performance. Please refer to Table 2 in the revised manuscript for detailed results.

---

### Official Review · Reviewer_q4E4 · 2025-11-01

**Soundness:** 3
**Presentation:** 4
**Contribution:** 2
**Rating:** 2
**Confidence:** 3

**Summary:**

This paper proposes PLDiv, a diversity metric for datasets based on persistence landscapes from topological data analysis (TDA). The authors argue that existing diversity measures, such as entropy- or kernel-based approaches, largely overlook geometric and structural properties of data. PLDiv quantifies diversity by integrating persistence landscapes derived from persistent homology, capturing both local and global topological features. The method is evaluated across synthetic, text, and image datasets, demonstrating advantages in interpretability, robustness, and geometry-awareness over baseline methods like Vendi Score, DCScore, and MAGAREA.

**Strengths:**

1.	Theoretical Grounding: The paper provides a closed-form expression for PLDiv and proves it satisfies key diversity axioms (effective size, monotonicity, twin property, symmetry), enhancing its credibility.
2.	Empirical Validation: Extensive experiments across modalities (synthetic, text, image) and tasks (subset selection, curvature regression, semantic diversity) show PLDiv’s consistency and superiority over competing methods.
3.	Interpretability: The connection between persistence lifetimes and diversity offers an intuitive and geometrically meaningful interpretation.

**Weaknesses:**

1.	The fundamental limitation of the proposed metric is that its geometric perspective on diversity may not align with human intuition, which greatly hinder its practical value. Please consider this example: in the experiment of Section 5.2, do texts generated with a higher temperature truly possess more meaningful diversity that mitigates the self-enforcing homogenization, which is the objective declared in the paper, or do they merely produce more variations of "LLM-ish" text?
2.	Computation Cost: Although sparse PLDiv is proposed, the full method remains computationally expensive compared to lighter baselines like Vendi Score or DCScore.

**Questions:**

1.	As the scale of training data is continuously increasing in practice, how does PLDiv scale to very large datasets in terms of cost, e.g. time and memory? Is it possible that the algorithmic complexity be explicitly stated?
2.	How sensitive is PLDiv to the choice of distance metric or embedding model, especially in semantic tasks where embedding quality varies significantly?
3.	As the Weakness 2 stated, there is a gap between the degree of geometric dispersion and semantically rich diversity in data or generated content. In this regard, what is your perspective?

---

> ### Author Response · Authors · 2025-11-24
>
> Thanks for your insightful comments. We provide our response here.
>
> >1. The fundamental limitation of the proposed metric is that its geometric perspective on diversity may not align with human intuition, which greatly hinder its practical value. Please consider this example: in the experiment of Section 5.2, do texts generated with a higher temperature truly possess more meaningful diversity that mitigates the self-enforcing homogenization, which is the objective declared in the paper, or do they merely produce more variations of "LLM-ish" text?
>
> We updated our manuscript to include a human-annotated dataset. The dataset from Tevet \& Berant (2021) contains a human-annotated diversity score, which we use as the ground truth in the revised Section 5.3, in addition to the softmax temperature. We observe a trend consistent with previous experiments. In addition to the correlation score, we also report R² and MSE. The results show that PLDiv outperforms alternative metrics across tasks and embedding models. This demonstrates the strong practical value of our proposed metric. Please refer to the updated Figures 4 and 9 and the revised discussion in Section 5.3 for details. We also added a visualization (Figure 10) in Appendix D.4 to illustrate that PLDiv remains robust across different distance matrices.
>
> >2. Computation Cost: Although sparse PLDiv is proposed, the full method remains computationally expensive compared to lighter baselines like Vendi Score or DCScore.
>
> Our primary contribution is the development of a reliable and highly accurate measure of dataset diversity. Although fast computation was not the initial focus, we propose a sparse PLDiv method that significantly improves computational efficiency with negligible error, making our approach both accurate and practically efficient. In our updated manuscript, we demonstrate that our metric is more effective than the Vendi Score for large-scale ImageNet-5k datasets under different sample sizes ranging from 5k to 40k. Across several experiments (Section 5.2-5.5), we observed that DCScore, while computationally efficient, demonstrates inferior performance in measuring diversity. Therefore, PLDiv achieves a strong balance between accuracy and efficiency.
>
> >3. As the scale of training data is continuously increasing in practice, how does PLDiv scale to very large datasets in terms of cost, e.g. time and memory? Is it possible that the algorithmic complexity be explicitly stated?
>
> We have revised Section 5.6 (Computational Complexity) to provide a more comprehensive comparison and analysis. Sample sizes of 5k, 10k, 20k, 30k, and 40k were drawn from ImageNet-1K, and experiments were conducted using ResNet-50 embeddings with cosine similarity/distance, each repeated five times to estimate uncertainty. Our sparse method performs comparably to other methods when the sample size is below 10k. However, as the sample size increases, even the standard PLDiv surpasses the Vendi Score in performance. Please refer to Table 2 in the revised manuscript for detailed results. The standard PLDiv requires $\mathcal{O}(n^2)$ time and memory to accommodate the full dense distance matrix, but the sparse PLDiv reduces the space complexity to linear $\mathcal{O}(n)$ (specifically $\mathcal{O}(C(\epsilon)n)$).
>
> >4.  How sensitive is PLDiv to the choice of distance metric or embedding model, especially in semantic tasks where embedding quality varies significantly?
>
> In Section 5.3, we have added two additional embedding models: Qwen3-Embedding-4B and Qwen3-Embedding-8B. As a result, the embedding models used in this experiment now span dimensions of 384, 768, 1024, 2560, and 4096. The revised Figures 4 and 9 show that our proposed PLDiv consistently outperforms existing metrics across varying embedding dimensions, and demonstrates particularly strong performance on the two recent, high-dimensional embedding models (Qwen3-4B and Qwen3-8B).
>
> To examine the sensitivity to the choice of distance matrix, we compare cosine distance and Euclidean distance on text embeddings. Our results show that, regardless of the choice of distance matrix, our metric consistently delivers reliable performance—ranking first in terms of $R^2$, MSE, and Pearson correlation. Please see Appendix D.4 and Figure 10.

---

> > ### Author Response · Authors · 2025-11-24
> >
> > >5. As Weakness 2 states, there is a gap between the degree of geometric dispersion and semantically rich diversity in data or generated content. In this regard, what is your perspective?
> >
> > We thank the reviewer for raising this insightful point. We believe the capability of an embedding space to represent semantically rich diversity is closely tied to its dimensionality. In our updated text embedding experiment (Section 5.3), we included three models corresponding to pre-LLM sentence embedding architectures (embedding dimensions ranging from 384 to 1024) and two models based on modern large language models with embedding dimensions of 2056 and 4096, respectively. The increased dimensionality of these latter models provides a higher-capacity geometric space, enabling better separation of distinct semantic concepts. Experimental results indicate that PLDiv performs more effectively on LLM-based, high-dimensional embedding models under both temperature and human-annotated evaluation settings, demonstrating the robustness and superiority of our proposed metric.
> >
> > Thanks again for your review. Given the subitem ratings and questions, we were quite surprised by the low overall score. If our responses have addressed your concerns, we kindly ask you to consider raising your rating. If you have any additional questions, we would be very happy to address them.

---

### Official Review · Reviewer_Eq6a · 2025-11-07

**Soundness:** 3
**Presentation:** 3
**Contribution:** 3
**Rating:** 6
**Confidence:** 4

**Summary:**

This paper proposes a topology-based diversity metric, PLDiv, which quantifies dataset diversity without relying on labels or reference sets. By integrating persistence landscapes over Vietoris–Rips filtrations, it captures both spatial dispersion and topological connectivity. Experiments on curvature point clouds, text, and image embeddings show that PLDiv effectively characterizes the structural properties of embedding distributions.

**Strengths:**

1. The paper introduces a novel perspective to dataset diversity measurement by integrating topological persistence with geometric embeddings.

2. The experiment results shows this paper effectively captures structural properties in both geometric point clouds (e.g., curvature data) and high-dimensional embedding spaces (e.g., textual and visual representations), demonstrating strong generality and adaptability across different data modalities.

3. The paper is generally clear and well written and the methodology is explained step by step with illustrative figures.

**Weaknesses:**

1. Given the potential computational and memory overhead of the topological process, it would be important to evaluate how the proposed method performs on large-scale datasets.

2. Although the method is applied to high-dimensional embeddings, the paper lacks a systematic analysis across different feature dimensions.

3. If the dataset follows a long-tailed or highly imbalanced distribution, it remains unclear how this diversity measure should be interpreted or whether it might be dominated by the majority clusters.

4. Unlike prior diversity metrics such as Vendi Score, DCScore, or MAGAREA, which include human-annotated or human-evaluated benchmarks to validate semantic diversity, this paper relies solely on embedding-level geometric correlations without external semantic validation.

5. Lack of discussions of theoretical or empirical analysis of space complexity, leaving the memory scalability of PLDiv unclear.

6. The authors could clarify the distinction between \epsilon (filtration) and \epsilon (sparsification rate).

7. The definition, proposition and remarks in Section 4 are numbered as (3.x) instead of (4.x).

**Questions:**

Please see weaknesses.

---

> ### Author Response · Authors · 2025-11-24
>
> Thank you for your effort in reviewing our work and for your thoughtful comments. Here are our responses.
>
> > 1. Given the potential computational and memory overhead of the topological process, it would be important to evaluate how the proposed method performs on large-scale datasets.
>
> We have revised Section 5.6 (Computational Complexity) to provide a more comprehensive comparison and analysis. Sample sizes of 5k, 10k, 20k, 30k, and 40k were drawn from ImageNet-1K, and experiments were conducted using ResNet-50 embeddings with cosine similarity/distance, each repeated five times to estimate uncertainty. Our sparse method performs comparably to other methods when the sample size is below 10k. However, as the sample size increases, even the standard PLDiv surpasses the Vendi Score in performance. Please refer to Table 2 in the revised manuscript for detailed results.
>
> >2. Although the method is applied to high-dimensional embeddings, the paper lacks a systematic analysis across different feature dimensions.
>
> Thank you for your comments. In Section 5.3 (Text Semantic Embeddings Experiment), we have added two additional embedding models: Qwen3-Embedding-4B and Qwen3-Embedding-8B. As a result, the embedding models used in this experiment now span dimensions of 384, 768, 1024, 2560, and 4096. The revised Figures 4 and 9 show that our proposed PLDiv consistently outperforms existing metrics across varying embedding dimensions, and demonstrates particularly strong performance on the two recent, high-dimensional embedding models (Qwen3-4B and Qwen3-8B). We also added a visualization (Figure 10) in Appendix D.4 to illustrate that PLDiv remains robust across different distance matrices.
>
> > 3. If the dataset follows a long-tailed or highly imbalanced distribution, it remains unclear how this diversity measure should be interpreted or whether it might be dominated by the majority clusters.
>
> Thank you for raising this concern. To address it, we use the toy example D4 (a single cluster containing 200 data points) as the baseline, and then gradually separate 20 data points at a time, scattering them randomly. This process forms five imbalanced datasets with one large cluster. PLDiv demonstrates that diversity increases monotonically, as shown in Figure 7 in Appendix D.1.1. Furthermore, we have added a more comprehensive experimental comparison in Section 5.5, which also addresses this concern.
>
> > 4. Unlike prior diversity metrics such as Vendi Score, DCScore, or MAGAREA, which include human-annotated or human-evaluated benchmarks to validate semantic diversity, this paper relies solely on embedding-level geometric correlations without external semantic validation
>
> We updated our manuscript to include this. The dataset from Tevet \& Berant (2021) contains a human-annotated diversity score, which we use as the ground truth in the revised Section 5.3, in addition to the softmax temperature. We observe a trend consistent with previous experiments. In addition to the correlation score, we also report R² and MSE. The results show that PLDiv outperforms alternative metrics across tasks and embedding models. Please refer to the updated Figures 4 and 9 and the revised discussion in Section 5.3 for details.
>
> > 5. Lack of discussions of theoretical or empirical analysis of space complexity, leaving the memory scalability of PLDiv unclear.
>
> We updated Section 5.6 (Computation Complexity). While standard PLDiv requires $\mathcal{O}(n^2)$ memory to accommodate the full dense distance matrix, Sparse PLDiv reduces the space complexity to linear $\mathcal{O}(n)$ (specifically $\mathcal{O}(C(\epsilon)n)$). This efficiency is achieved by utilizing the geometric sparsification scheme to construct a Minimum Spanning Tree (MST) with significantly fewer edges, combined with a closed-form expression that calculates PLDiv directly from the MST weights.
>
> > 6. The authors could clarify the distinction between $\epsilon$ (filtration) and $\epsilon$ (sparsification rate)
>
> Thanks for this comment. We fixed it by changing the filtration radius to $r$.
>
> > 7. The definition, proposition and remarks in Section 4 are numbered as (3.x) instead of (4.x).
>
> Thanks for pointing it out. We have fixed this topo.

---

### Author Response · Authors · 2025-11-24
**General Response**

We sincerely thank all reviewers for their thoughtful and constructive feedback, which allows us to improve our paper. Below, we summarize our key contributions, the common concerns raised across reviewers, and the major updates made in the revised version.

**Our contributions:** Our work introduces PLDiv, a mathematically rigorous and geometry-aware diversity metric derived from persistent landscapes of topological data analysis. PLDiv bridges the gap between geometric structure and statistical diversity. Both theoretically and empirically, we show that PLDiv satisfies four core diversity axioms. We provide a closed-form, interpretable expression connecting topological persistence with distributional variance. To validate the advantages of PLDiv, we conducted experiments across multiple modalities.

**Common concerns and our revisions:**
> 1. Computational efficiency and scalability (raised by Reviewers Eq6a, q4E4, jSSq, Qxac, and 62tN)

We updated Section 5.6 (Computational Complexity) with experiments on ImageNet-1K (sample sizes 5k–40k). By utilizing closed-form expressions of PLDiv and Minimum Spanning Tree, the sparse PLDiv achieves linear memory growth and remains efficient for larger datasets while maintaining accuracy. We observed that PLDiv becomes more computationally efficient than Vendi Score for sample sizes larger than 20,000. Results are summarized in Table 2 of the revised manuscript.

> 2. Validation against human-annotated diversity score (raised by Reviewers Eq6a, q4E4, and 62tN)

As suggested, we incorporated the Tevet \& Berant (2021) dataset containing human-annotated diversity scores as ground truth. Moreover, we added two recent LLM-based high-dimensional embedding models (Qwen-Embedding-4B and Qwen-Embedding-8B). The updated Section 5.3 and Figures 4 and 9 now report Pearson correlation, R², and MSE, demonstrating that PLDiv aligns more closely with both the softmax temperature and human judgments across embedding models than any other diversity metrics. Figure 10 further illustrates the impact of different distance matrices and kernels on metric performance, demonstrating that PLDiv consistently achieves superior results in most scenarios.

> 3. Rigorous comparison with baselines and Geometry-Awareness justification (raised by Reviewers Eq6a, Qxac, 62tN)

We added Section 5.5 and a new experiment using parameterized geometric functions (Figure 6) to test "geometry-awareness." We show that existing metrics often fail to distinguish between distinct geometric structures that share similar dispersion. PLDiv effectively captures these geometric (shape) differences. We also added quantitative metrics $R^2$, MSE, and Pearson correlation to demonstrate the advantage of PLDiv in Section 5.3.

> 4. Reproducibility and implementation details (raised by the Reviewer Qxac)

We added detailed implementation settings and parameter configurations in Appendix D, and released our code in the supplementary material to ensure full reproducibility.

> 5. Discussion of limitations (raised by the Reviewer 62tN)

We introduced a Limitations Section (Appendix E), discussing scenarios where PLDiv may be less reliable, and outlined potential future directions, such as alternative filtrations.

We appreciate all your efforts in reviewing our work. All revisions in the manuscript are highlighted in blue. We hope these revisions and the detailed individual responses below have addressed your concerns.

---

### Meta-Review · Area_Chair_MXP2 · 2026-01-08

**Summary:**

The reviews are high-quality and provide a comprehensive evaluation of the proposed method.  The reviewers have questioned the novelty, usefulness, and experimental scale of the article. The authors have done a great job and presented new results for almost all the points raised. However, the core criticisms remain, such as the distinction from existing work and high computational costs.

Unfortunately, in the reviews, I also do not see a serious misunderstanding whose removal would considerably increase the ratings.

**Reviewer Concerns:**

Eq6a
- Notes the lack of experiments on large datasets.
- Missing space complexity
- Mentions the uncertainty in explaining the measure in imbalanced and long-tailed data.

q4E4
- Asks, will the measure align with human intuition? The authors introduce a new ground-truth dataset to support their claims.
- Computational costs are high. The authors prepare a new scalability experiment to argue that the method is scalable.

jSSq
- Mentions a lack of novelty and a weak theoretical contribution. Also notes computational costs. The authors disagree with the reviewer and argue to the contrary.

Qxac
- In a comprehensive review, the reviewer raises questions about the scalability, novelty, and overall usefulness of the proposed method. The authors have prepared a lengthy response to the reviewer and attempted to address every criticism.

62tN
- Notes the lack of large validations. The authors present new results.
- Argues that the method's results are not novel. Also raises doubts on computational costs. The authors prepare a new scalability analysis, but the authors note that the method does not outperform the baselines for large sample sizes.

**Reviewer Scores:**

Eq6a rates 6 and would keep the score
q4E4 rates 2, and would likely increase to 4
qxAc rates 4 and would likely keep the score.
62tN rates 4 nd would most likely keep the score.

---

### Decision · Program_Chairs · 2026-01-26

Reject